# Realizable Bayes-Consistency for General Metric Losses

**Dan Tsir Cohen** [1]  **Steve Hanneke** [2]  **Aryeh Kontorovich** [1]

## Abstract

We study strong universal Bayes-consistency in the realizable setting for learning with general metric losses, extending classical characterizations beyond 0-1 classification (Bousquet et al., 2021; Hanneke et al., 2021) and real-valued regression (Attias et al., 2024). Given an instance space $(\mathcal{X}, \rho)$, a label space $(\mathcal{Y}, \ell)$ with possibly unbounded loss, and a hypothesis class $\mathcal{H} \subseteq \mathcal{Y}^{\mathcal{X}}$, we resolve the realizable case of an open problem presented in Tsir Cohen & Kontorovich (2022). Specifically, we find the necessary and sufficient conditions on the hypothesis class $\mathcal{H}$ under which there exists a distribution-free learning rule whose risk converges almost surely to the best-in-class risk (which is zero) for every realizable data-generating distribution. Our main contribution is this sharp characterization in terms of a combinatorial obstruction: Similarly to Attias et al. (2023), we introduce the notion of an infinite non-decreasing $(\gamma_k)$-Littlestone tree, where $\gamma_k \to \infty$. This extends the Littlestone tree structure used in Bousquet et al. (2021) to the metric loss setting.

## 1. Introduction

Many learning problems are naturally expressed with a *metric loss* on the label space: one predicts a label $y \in \mathcal{Y}$ and is penalized by the distance $\ell(\hat{y}, y)$. This encompasses multi-class prediction (discrete metrics), regression (absolute loss), structured outputs with edit-type losses, and general cost-sensitive prediction. In such settings $\mathcal{Y}$ is typically infinite and $\ell$ may be unbounded. While unbounded losses are standard in regression, they are far less understood in the broader metric-loss setting from the most basic distribution-free perspective: *when is strong universal consistency possible at all?*

[1]Data Science Research Center, Ben-Gurion University of the Negev [2]Purdue University, USA. Correspondence to: Dan Tsir Cohen <dantsir@post.bgu.ac.il>.

*Proceedings of the 43rd International Conference on Machine Learning*, Seoul, South Korea. PMLR 306, 2026. Copyright 2026 by the author(s).

**Realizable strong universal Bayes-consistency.** Fix an instance space $(\mathcal{X}, \rho)$, a label space $(\mathcal{Y}, \ell)$ (where $\rho$ and $\ell$ are metrics), and a hypothesis class $\mathcal{H} \subseteq \mathcal{Y}^{\mathcal{X}}$. $\ell$ is the (possibly unbounded) loss function. Given an i.i.d. sample $(X_i, Y_i)_{i \in [n]} \sim \bar{\mu}^n$ from an unknown distribution $\bar{\mu}$ on $\mathcal{X} \times \mathcal{Y}$, a learner outputs a predictor $f_n : \mathcal{X} \to \mathcal{Y}$ and incurs risk

$$R_{\bar{\mu}}(f_n) := \mathop{\mathbb{E}}_{(X,Y) \sim \bar{\mu}} \ell(f_n(X), Y).$$

We focus on the *realizable* setting: $\bar{\mu}$ is realizable by $\mathcal{H}$ if $\inf_{h \in \mathcal{H}} R_{\bar{\mu}}(h) = 0$.[1] We ask for a *distribution-free* learning rule whose risk converges to $0$ *almost surely* for *every* realizable $\bar{\mu}$. This is the natural strong-law analogue of universal consistency. In bounded loss settings, realizability is already a powerful structural assumption and often collapses the distinction between "risk goes to $0$" and "errors go to $0$." For unbounded metric losses, this intuition fails in a very concrete way: a learner may make mistakes on events whose probability decays with $n$, yet the *scale* of those mistakes can grow so quickly that the risk does not vanish (and can even be infinite). Understanding when such catastrophic rare-event failures are *avoidable uniformly over all realizable distributions* is the crux of the problem.

**Why unbounded metric losses are genuinely harder.** With any bounded loss, every mistake costs at most a constant, so controlling the probability of error is essentially synonymous with controlling the risk. With general metric losses, a learner can be forced to *guess between two labels that are extremely far apart* on regions of the instance space that are rare but not ignorable. Realizability alone does not preclude this: an adversary can "hide" an infinite sequence of increasingly separated label choices behind a rapidly decaying probability mass. This phenomenon is absent in the bounded-loss theory and is not captured by purely distributional finiteness conditions (e.g., candidate assumptions such as $R^\star < \infty$); indeed, in Section 3 we give an explicit counterexample showing that such finiteness-type conditions do not suffice to guarantee any distribution-free consistent learner in the realizable unbounded-loss regime.

---

[1]Under the compact-parameterization assumption used in our main results (compact $\Theta$, $h$ continuous in $\theta$, Lemma 4.3), realizability is equivalent to the existence of $f^\star \in \mathcal{H}$ with $Y = f^\star(X)$ almost surely (strict realizability), since the infimum over the compact image of $\Theta$ is attained. We use this equivalence freely in proofs. See Appendix A.4.5 for further discussion.

**Main result (informal; refer to Section 4).** Our contribution is a sharp hypothesis-class characterization of when realizable strong universal Bayes-consistency is possible for general metric losses. The characterization is in terms of a single combinatorial obstruction: an *unbounded-gap* Littlestone tree.

Informally, an infinite non-decreasing $(\gamma_k)$-Littlestone tree is a full binary tree of instances whose depth-$k$ nodes admit two outgoing labels at metric distance at least $\gamma_k$, where $\gamma_k \uparrow \infty$, and such that every finite root-to-leaf labeling pattern is realizable by some $h \in \mathcal{H}$ (Definition 4.1). Under a mild compact-parameterization assumption (Lemma 4.3), this finite-prefix condition automatically ensures that every *infinite* path is realized by a single hypothesis. In particular, the counterexample of Section 3 naturally satisfies this assumption, demonstrating that unlearnability remains possible even under it (see Appendix A.4.3 for details).

*For metric losses (possibly unbounded), realizable strong universal Bayes-consistency is possible for $\mathcal{H}$ if and only if $\mathcal{H}$ does **not** contain an infinite non-decreasing $(\gamma_k)$-Littlestone tree with $\gamma_k \to \infty$.* Moreover:

- (**Lower bound.**) If such a tree exists, then for *every* learning algorithm $\mathcal{A}$ there is a realizable distribution $\bar{\mu}$ for which the learner suffers catastrophic rare-event errors; in fact, $\mathbb{E}_{S_n \sim \bar{\mu}^n}[R_{\bar{\mu}}(\mathcal{A}(S_n))] = \infty$ for every $n$ (Theorem 5.1).

- (**Upper bound.**) If no such tree exists, then there is an explicit distribution-free learning rule whose risk converges to $0$ almost surely for every realizable $\bar{\mu}$ (Theorem 6.5).

**Proof architecture.** The lower bound converts an unbounded-gap tree into a realizable distribution on a random root-to-leaf path: at depths the sample misses (which exist for every finite $n$), the learner is forced to guess between two labels separated by $\gamma_k$, and if $\gamma_k$ grows fast enough the expected risk diverges. This is a metric-loss analogue of the classical adversarial role of Littlestone trees, but with the new ingredient that the *scale* of the ambiguity diverges. The upper bound takes a complementary game-theoretic route. Absence of an infinite unbounded-gap tree implies the learner wins an infinite Gale-Stewart game over realizable adversarial extensions, in the sense of Bousquet et al. (2021). Simulating a winning strategy on the i.i.d. sample stabilizes a.s. after finitely many advances, yielding for each $x$ a bounded-diameter set of admissible labels that contains the true label. A data-dependent countable partition of $\mathcal{X}$ then reduces the problem to countably many bounded-loss subproblems handled by existing metric-loss learners.

**Contributions.** In summary, this paper provides:

- a **complete hypothesis-class characterization** of realizable strong universal Bayes-consistency for general metric losses, via the infinite unbounded-gap Littlestone tree obstruction (Theorem 4.5);

- a **constructive learning rule** in the obstruction-free regime, proven to be strongly universally Bayes-consistent in the realizable setting (Theorem 6.5);

- a **separation** demonstrating that natural finiteness-type conditions (e.g. candidates based on $R^\star < \infty$) do not capture the correct characterization in the unbounded metric-loss realizable setting, resolving the realizable case of the open problem posed by Tsir Cohen & Kontorovich (2022).

Concrete examples of hypothesis classes that do and do not admit an unbounded-gap tree, illustrating the practical scope of the characterization, are presented in Appendix A.1. Extending the characterization to the *agnostic* setting, where $\bar{\mu}$ need not be realizable, is an important direction for future work; we discuss the barriers to such an extension in Appendix A.5.

**Organization.** Section 2 discusses related work, with emphasis on universal learning and tree/game characterizations (Bousquet et al., 2021; Hanneke et al., 2023; Attias et al., 2024), metric-loss learning algorithms (Tsir Cohen & Kontorovich, 2022), and combinatorial characterizations of learnability (Brukhim et al., 2022). Section 3 presents a counterexample showing that finiteness-type conditions alone do not guarantee distribution-free consistency for unbounded metric losses. Section 4 presents the main characterization and its proof, which consists of two parts: Section 5 proves the realizable lower bound: the existence of an infinite unbounded-gap Littlestone tree implies that no learner can be consistent (Theorem 5.1). Section 6 proves the matching upper bound: in the absence of such a tree, we construct a strongly universally Bayes-consistent learner (Theorem 6.5).

## 2. Related work

**Universal learning and game-theoretic characterizations.** Our work builds on the recent program of understanding *universal* (i.e., distribution-free) learnability through the lens of infinite games and combinatorial trees. In the realizable 0-1 setting, Bousquet et al. (2021) introduced the universal learning model (allowing distribution-dependent constants) and developed a sharp theory of achievable rates via Gale-Stewart games and tree obstructions, highlighting the role of infinite Littlestone-type structures and measurability of

winning strategies. This framework was extended to multiclass prediction by Hanneke et al. (2023), who identified additional tree-based obstructions governing the universal rate landscape beyond the binary setting. More recently, Attias et al. (2023) studied realizable regression in both PAC and online learning settings (notably for cut-off and absolute losses), providing characterizations of learnability via multiple combinatorial dimensions: scaled Graph dimension for ERM learnability, scaled One-Inclusion-Graph (OIG) dimension for general PAC learnability, and an online dimension based on *scaled* Littlestone trees (where the label separation is controlled by a resolution parameter or a scale sequence) that characterizes optimal cumulative loss in online learning (resolving an open question from prior work). The present paper continues this line in a different direction: rather than focusing on rates under bounded losses, we study *strong* universal Bayes-consistency under *general metric losses*, including unbounded ones, and provide a necessary-and-sufficient characterization in terms of an *unbounded-gap* Littlestone obstruction.

**Learning with metric losses and unbounded risk.**
Learning with general metric losses has recently attracted attention as a common generalization of multiclass classification and real-valued regression. Most directly related is the MedNet algorithm of Tsir Cohen & Kontorovich (2022), which gives a constructive strong universal Bayes-consistent learner in the *agnostic* setting under separability assumptions and a *bounded-in-expectation* (BIE) condition on $(\mathcal{Y}, \ell)$. That work extends the "optimistically universal" nearest-neighbor paradigm initiated for multiclass 0-1 learning on metric instance spaces (e.g., OptiNet and follow-ups Hanneke et al., 2021) to general metric label spaces, and emphasizes that naïve extensions of classical metric-space methods can fail under general loss geometries. Our results are complementary in two ways. First, we focus on the *realizable* setting and ask when strong universal Bayes-consistency is possible *as a function of the hypothesis class* $\mathcal{H}$, rather than imposing integrability conditions on $\mathcal{Y}$ or distributional conditions on $\bar{\mu}$. Second, the lower-bound construction underlying our characterization clarifies why unbounded losses fundamentally change the picture: even when the Bayes optimal risk is 0, rare mispredictions at increasing loss scales can preclude strong universal Bayes-consistency. In particular, our counterexample shows that finiteness-type conditions such as $R^* < \infty$, suggested as a candidate in Tsir Cohen & Kontorovich (2022), are not sufficient in general without further structural assumptions.

**Metric-space learning, Lipschitz methods, and doubling structure.** A parallel thread studies learning and generalization when instances are endowed with a metric and algorithms must leverage only distance information. A prominent approach models classifiers/regressors as Lipschitz functions and controls complexity via metric notions such as (doubling) dimension and fat-shattering. For example, Gottlieb and Krauthgamer and coauthors developed efficient large-margin classification algorithms for metric data whose runtime and guarantees depend on the doubling dimension and approximate proximity search/Lipschitz extension primitives (e.g., Gottlieb et al., 2014; see also proximity data structures in nearly doubling metrics Gottlieb & Krauthgamer, 2013). These works provide algorithmic and finite-sample perspectives tailored to geometric regimes (bounded diameter, doubling, or margin assumptions). Our setting is orthogonal: we do not assume low-dimensional geometry or bounded diameter globally, and the obstruction we identify is *combinatorial* (tree-based) rather than geometric. That said, our *upper-bound* construction explicitly exploits bounded-diameter subproblems (obtained via a data-driven partition of $\mathcal{X}$) and can therefore be viewed as a reduction that makes it possible to plug in existing bounded-loss or bounded-range metric learners (including MedNet as used here) locally.

**Connections to classical combinatorial dimensions.**
Tree-based parameters are ubiquitous across learning models. In the uniform/PAC theory, VC dimension (binary) and its multiclass extensions (e.g., Natarajan and related dimensions) characterize learnability, with sharp recent results such as the DS-dimension characterization of multiclass PAC learnability (Brukhim et al., 2022). In adversarial online learning, the Littlestone dimension and its multiclass generalizations govern mistake bounds and regret. The universal-learning line (Bousquet et al., 2021; Hanneke et al., 2023; Attias et al., 2024) reveals that *infinite* tree obstructions (and refinements thereof) also control distribution-dependent behavior beyond uniform convergence. Our contribution is to show that, for general *metric* losses in the *realizable* setting, the decisive obstruction is the existence of an infinite Littlestone-type tree whose label gaps can be forced to grow without bound along depth. This pinpoints the precise way in which unbounded loss magnitudes interact with realizability to make strong universal Bayes-consistency either possible or impossible.

## 3. $R^* < \infty$ is insufficient

In Tsir Cohen & Kontorovich (2022), a strongly universally Bayes-consistent learner for unbounded losses (MedNet) is achieved only when the label space $\mathcal{Y}$ is BIE. The label space $\mathcal{Y}$ is *bounded in expectation* (BIE) if $\mathbb{E}_{(X,Y) \sim \bar{\mu}} \ell(y_0, Y) < \infty$ for some $y_0 \in \mathcal{Y}$.

The BIE condition is a natural metric analogue of the real-valued moment assumption $\mathbb{E}|Y| < \infty$. It was shown to be sufficient for Bayes consistency of MedNet, and it motivates the broader question of what, precisely, is required

for strong Bayes-consistency when the loss is unbounded. Accordingly, Tsir Cohen & Kontorovich (2022) posed the open problem of giving a necessary and sufficient condition on $(\mathcal{X}, \rho)$, $(\mathcal{Y}, \ell)$, and $\bar{\mu}$ under which MedNet- or some other learning algorithm - is strongly Bayes-consistent. A natural and optimistic candidate was the finiteness of the Bayes risk, $R^* < \infty$. In this section we show that this condition is not sufficient: without further assumptions, no strongly universally Bayes-consistent learner can exist.

We will show that for *any* learning algorithm $\mathcal{A}$, there *exists* a distribution $\mu$, such that the expected risk (over the random given sample $S_n \sim \mu^n$) of the learned hypothesis $h_n := \mathcal{A}(S_n)$ does *not* go to zero as sample size grows to $\infty$:

$$\mathbb{E}_{S_n \sim \mu^n} [R(\mathcal{A}(S_n))] \not\to 0 \text{ as } n \to \infty$$

Meaning that without further assumptions - no learner is strongly universally Bayes-consistent. Let $\mathcal{X}$ be the open interval $(0, 1)$. Let $\mathcal{Y} = \mathbb{N}_0 := \{0, 1, 2, \dots\}$, with the standard $\ell_1$ loss. Define for any $k \in \mathbb{N}$ :

$$I_k := \left( \frac{1}{2^k}, \frac{1}{2^{k-1}} \right).$$

Thus $\mathcal{X} = \bigcup_{k \in \mathbb{N}} I_k$. We look at the realizable case, with the uniform distribution $\mu$ over $\mathcal{X}$, realized by the family of functions:

$$\mathcal{H} := \left\{ f : \forall k \in \mathbb{N}, \forall x \in I_k : f(x) \in \{0, 2^{2k+1}\} \right\},$$

thus defining the family of possible distributions on $(\mathcal{X}, \mathcal{Y})$. We now define a *realizable* distribution via an explicit product (coin-flip) construction. Let $B = (B_k)_{k \geq 1}$ be i.i.d. fair bits, $B_k \sim \text{Ber}(1/2)$, and define $f_B \in \mathcal{H}$ by

$$f_B(x) := \begin{cases} 0 & x \in I_k \text{ and } B_k = 0, \\ 2^{2k+1} & x \in I_k \text{ and } B_k = 1. \end{cases}$$

For each fixed $B$, let $\bar{\mu}_B$ be the distribution on $\mathcal{X} \times \mathcal{Y}$ given by $X \sim \text{Unif}(0, 1)$ and $Y = f_B(X)$. Then $\bar{\mu}_B$ is realizable by $f_B$, hence $R^*(\bar{\mu}_B) = 0 < \infty$. Thus, we observe this as a problem where the learner (learning algorithm) sees an i.i.d. sample $S_n = \{(X_1, Y_1), \dots, (X_n, Y_n)\} \sim \bar{\mu}_B^n$ and aims to output a predictor $h_n := \mathcal{A}(S_n)$ so as to minimize the risk:

$$R_{\bar{\mu}_B}(h_n) := \mathbb{E}_{(X,Y) \sim \bar{\mu}_B} \ell(h_n(X), Y).$$

The key point is that, conditional on any realized sample $S_n$, infinitely many intervals $I_k$ remain unobserved, so $B_k$ stays a fresh fair coin on each such interval. On $I_k$, at least one of the two possible labels $\{0, 2^{2k+1}\}$ must incur loss at least $2^k$ (by the triangle inequality and integration over $|I_k| = 2^{-k}$), so with conditional probability at least $1/2$ the interval contributes $\geq 2^k$ to the risk. Independence across unseen $k$ and Borel-Cantelli then force infinitely many such contributions, implying $R_{\bar{\mu}_B}(h_n) = \infty$ almost surely. The full derivation is in Appendix A.6.

# 4. Main result: characterization of $\mathcal{H}$

We now state the main theorem, which characterizes when strong universal Bayes-consistency is achievable in the realizable setting under general (possibly unbounded) metric losses. The characterization is in terms of a single combinatorial obstruction: an *unbounded-gap* Littlestone tree.

**Definition 4.1** (non-decreasing-$\gamma_k$-Littlestone tree). Fix a non-decreasing sequence $(\gamma_k)_{k \geq 1}$ with $\gamma_k \to \infty$. A *non-decreasing-$\gamma_k$-Littlestone tree for $\mathcal{H}$* (a metric-loss variant of the classical Littlestone tree (Littlestone, 1988); see also Bousquet et al. (2021, Definition 1.7)) is a rooted full binary tree whose internal nodes at depth $k$ are labeled by instances $x_{k,i} \in \mathcal{X}$, and whose two outgoing edges from each such node are labeled by $y_{k,i,1}, y_{k,i,2} \in \mathcal{Y}$ satisfying $\ell(y_{k,i,1}, y_{k,i,2}) \geq \gamma_k$, with the property that every *finite* root-to-leaf path is realizable by some $h \in \mathcal{H}$ (i.e., $h$ matches the edge labels along that finite path).

**Definition 4.2** (realizable non-decreasing-$\gamma_k$-Littlestone tree). An *infinite non-decreasing-$\gamma_k$-Littlestone tree* is a tree of infinite depth satisfying the conditions of Definition 4.1 at every finite depth $k$. Such an infinite tree is *realizable* if, in addition to finite-prefix realizability, every infinite root-to-leaf path has its entire labeling realized by a *single* hypothesis in $\mathcal{H}$ (i.e., for each infinite path there exists $h \in \mathcal{H}$ matching all edge labels along that path).

**Lemma 4.3** (Bridging lemma: finite-prefix realizability implies infinite-path realizability). *Suppose $\mathcal{H}$ satisfies the measurability condition of Bousquet et al. (2020, Definition 3.3) with a **compact** parameter space $\Theta$ and a function $h : \Theta \times \mathcal{X} \to \mathcal{Y}$ that is **continuous in** $\theta$ (i.e., $\theta \mapsto h(\theta, x)$ is continuous for each fixed $x \in \mathcal{X}$). Then every infinite non-decreasing-$\gamma_k$-Littlestone tree (Definition 4.1) is automatically realizable (Definition 4.2).*

*Proof.* The proof follows from the finite-intersection property of compact spaces; see Appendix A.4. $\square$

*Remark* 4.4. Lemma 4.3 requires the compact-parameterization assumption (compact $\Theta$, $h$ continuous in $\theta$); we discuss in Appendix A.4 why this assumption is necessary in general, give a concrete example where the two tree notions differ without it, and argue that the assumption is mild and covers virtually all settings of practical interest.

**Theorem 4.5** (Main characterization). *Assume $(\mathcal{X}, \rho)$ and $(\mathcal{Y}, \ell)$ are Polish, and $\mathcal{H}$ satisfies the measurability condition of Bousquet et al. (2020, Definition 3.3) with compact $\Theta$ and $h$ continuous in $\theta$ (cf. Lemma 4.3). Then the following are equivalent:*

1. *There exists a (distribution-free) learning rule $\mathcal{A}$ such that for every realizable distribution $\bar{\mu}$ on $\mathcal{X} \times \mathcal{Y}$, the learned predictors $h_n := \mathcal{A}(S_n)$ satisfy*

   $$R_{\bar{\mu}}(h_n) \to 0 \qquad \text{almost surely as } n \to \infty.$$

2. $\mathcal{H}$ *admits no infinite non-decreasing-$(\gamma_k)$-Littlestone tree (for any non-decreasing gap sequence $(\gamma_k)$ with $\gamma_k \to \infty$).*

*Proof.* (1) $\implies$ (2): Contrapositive: suppose an infinite non-decreasing-$\gamma_k$-Littlestone tree exists (Definition 4.1). By Lemma 4.3, the tree is automatically realizable (Definition 4.2). Theorem 5.1 then shows that no learner can be consistent. (2) $\implies$ (1): If no such obstruction exists, then the explicit learner constructed in Section 6.5 is strongly universally Bayes-consistent for every realizable $\bar{\mu}$ (Theorem 6.5). $\qquad\square$

## 5. Lower bound: Existence of an infinite tree implies no consistency

**Theorem 5.1** (Lower bound). *Let $\mathcal{H}$ be a hypothesis class that contains a realizable infinite non-decreasing-$\gamma_k$-Littlestone tree with $\gamma_k \to \infty$. Then, for any learning algorithm $\mathcal{A}$, there exists a realizable distribution $\bar{\mu}$ such that the expected risk is infinite:*

$$\mathbb{E}_{S_n \sim \bar{\mu}^n}[R(\mathcal{A}(S_n))] = \infty$$

*for all $n$. In particular, no learner is consistent.*

*Proof sketch; full proof in Appendix A.7.* The argument generalizes the counterexample of Section 3: we construct a realizable distribution supported on selected tree nodes where the gap is large, then show every learner suffers infinite risk via a Borel-Cantelli argument on unseen nodes.

**Distribution construction.** Since $\gamma_k \to \infty$, we can select depths $k_1 < k_2 < \cdots$ along the tree where $\gamma_{k_m} \geq m^2$. Assign probability mass $p_m \propto 1/m^2$ to the depth-$k_m$ node (normalized so that $\sum p_m = 1$). A random root-to-leaf path is drawn by tossing independent fair coins $(B_k)_{k \geq 1}$, one per tree level; the coin $B_{k_m}$ selects which of the two outgoing labels at depth $k_m$ is the true label. Because the tree is realizable (Definition 4.2), the resulting distribution $\bar{\mu}_B$ is realizable by some $f_B^\star \in \mathcal{H}$.

**Unseen depths act as fair coins.** Fix any learner $\mathcal{A}$ and sample size $n$. A sample $S_n \sim \bar{\mu}_B^n$ can touch at most $n$ of the countably many depths $k_m$, so all but finitely many remain unobserved. At each unseen depth $k_m$, the coin $B_{k_m}$ is independent of $S_n$, meaning the learner's prediction $h_n(x_{k_m})$ is chosen with no information about the true label. Since the two candidate labels are at distance $\geq \gamma_{k_m} \geq m^2$, the triangle inequality forces $\mathbb{E}[\ell(h_n(x_{k_m}), Y) \mid S_n] \geq m^2/2$, regardless of what the learner predicts.

**Divergence via Borel-Cantelli.** More precisely, whichever label the learner's prediction is closer to, the "bad" coin outcome (the other label) occurs with conditional probability $1/2$, independently across unseen depths. On that event the loss contribution is at least $p_m \cdot m^2/2 = \Theta(1)$. By the second Borel-Cantelli lemma, infinitely many such bad events occur almost surely, so $R_{\bar{\mu}_B}(h_n) = \infty$ a.s. A Fubini argument over the random path $B$ and a countable intersection over $n \in \mathbb{N}$ then yield, for the fixed learner $\mathcal{A}$, a single distribution $\bar{\mu}$ with infinite expected risk at every sample size. $\qquad\square$

## 6. Upper bound: absence of an infinite tree implies strong universal consistency

In this section we complete the argument that, under a suitable combinatorial condition on $\mathcal{H}$, strong universal Bayes-consistent learning is possible in the realizable case, even when the loss is unbounded.

**Proof roadmap.** The upper bound proceeds in four steps: (1) we encode the tree obstruction as a Gale-Stewart game and invoke the measurable-strategy machinery of Bousquet et al. (2021) (Section 6.1); (2) we show that driving this game on an i.i.d. sample causes it to *stabilize* after finitely many rounds a.s., producing for each $x$ a set of admissible labels $H_K(x)$ of bounded diameter (Sections 6.2-6.3); (3) we partition $\mathcal{X}$ into countably many cells, each with a common bounded label region (Section 6.4); (4) on each cell we apply the bounded-range learner MedNet of Tsir Cohen & Kontorovich (2022) and aggregate (Sections 6.5-6.6).

**Standing measurability assumptions.** To avoid measure-theoretic pathologies (measurable determinacy / measurable strategies and measurable selections), we assume that both $(\mathcal{X}, \rho)$ and $(\mathcal{Y}, \ell)$ are Polish spaces. In particular, $(\mathcal{Y}, \ell)$ is separable, hence admits a countable dense subset, which we will use to build a countable partition of $\mathcal{X}$. In addition, we assume the hypothesis class $\mathcal{H}$ satisfies the measurability condition of Bousquet et al. (2020, Definition 3.3): there exists a Polish space $\Theta$ (the *parameter space*) and a Borel-measurable *evaluation map* $h: \Theta \times \mathcal{X} \to \mathcal{Y}$, $(\theta, x) \mapsto h(\theta, x)$, such that $\mathcal{H} = \{h(\theta, \cdot) : \theta \in \Theta\}$. We strengthen this by requiring $\Theta$ to be **compact** and $h$ to be **continuous in** $\theta$ (i.e., $\theta \mapsto h(\theta, x)$ is continuous for each fixed $x \in \mathcal{X}$; cf. Lemma 4.3, and see Appendix A.4 for a discussion of why this strengthening is needed and why it is mild), so that the game in Section 6.1 admits a *universally measurable* winning strategy for the learner. Whenever we define set-valued maps or indices using this strategy (e.g. $H_k(\cdot)$, $j_k(\cdot)$, and the cells $C_{k,j}$), we tacitly work with universally measurable versions so that these objects (and the final predictor) can be taken universally measurable.[2]

---

[2] Concretely, this can be ensured by standard selection results on Polish spaces (e.g. Jankov-von Neumann type selection) applied

### 6.1. A Gale-Stewart game

Fix a non-decreasing sequence $(\gamma_k)_{k \geq 1}$ with $\gamma_k \to \infty$. The particular choice of diverging gap schedule is inessential: if $\mathcal{H}$ contains an infinite non-decreasing-gap tree whose gaps tend to $\infty$, then for any prescribed diverging sequence $(\tilde{\gamma}_m)_{m \geq 1}$ one can obtain an infinite non-decreasing-$(\tilde{\gamma}_m)$ tree by passing to a subsequence of depths along which the original gaps exceed $\tilde{\gamma}_m$ ("skipping levels"). Consequently, the absence of an infinite non-decreasing-$(\gamma_k)$ tree for *some* diverging $(\gamma_k)$ is equivalent to the absence of any infinite unbounded-gap Littlestone tree obstruction.

Now, we define an infinite Gale-Stewart game between an adversary $P_A$ and a learner $P_L$. At round $k = 1, 2, \ldots$:

- $P_A$ chooses a triple $(\xi_k, \eta_{k,1}, \eta_{k,2}) \in \mathcal{X} \times \mathcal{Y} \times \mathcal{Y}$ such that $\ell(\eta_{k,1}, \eta_{k,2}) \geq \gamma_k$ and such that there exists an $h \in \mathcal{H}$ consistent with the previous play and satisfying $h(\xi_k) \in \{\eta_{k,1}, \eta_{k,2}\}$.

- $P_L$ chooses one of the two labels, denoted $\hat{\eta}_k \in \{\eta_{k,1}, \eta_{k,2}\}$.

The learner wins if at some finite round the adversary has no legal move; otherwise the adversary wins (i.e., if play continues forever). Recall the definition of a non-decreasing-$\gamma_k$-Littlestone tree from Definition 4.1. It is immediate that there exists an infinite non-decreasing-$\gamma_k$ Littlestone tree iff $P_A$ has a winning strategy. Hence, if no such infinite tree exists, then $P_L$ has a winning strategy. Under the Polish assumptions, we fix a *measurable* winning strategy $\sigma$ for $P_L$. The existence of such a measurable strategy is explained in Appendix A.3, where we verify that the conditions of Bousquet et al. (2020, Theorem B.1) are met by our game.

### 6.2. History-conditional label sets

**Definition 6.1** (History-conditional label sets). Let $\tau_k$ denote a finite history arising from a legal play under $\sigma$. Given such a history and a point $x \in \mathcal{X}$, define the *history-conditional feasible label set*

$$V_k(x) := \{y \in \mathcal{Y} : \exists h \in \mathcal{H}$$
$$\text{consistent with } \tau_k \text{ such that } h(x) = y\},$$

and the *history-conditional label set*

$$H_k(x) := \big\{y \in V_k(x) : \forall y' \in V_k(x) \text{ with } \ell(y, y') \geq \gamma_{k+1},$$
$$\sigma \text{ does not choose } y \text{ at round } k+1$$
$$\text{when presented with}$$
$$(\xi_{k+1}, \eta_{k+1,1}, \eta_{k+1,2}) = (x, y, y')\big\}.$$

_______________
to the analytic sets arising from the legal-move relation and the strategy $\sigma$.

Intuitively, $H_k(x)$ is the set of labels at $x$ that the strategy $\sigma$ *never* selects at the next round when paired with any sufficiently far *feasible* alternative.

**Lemma 6.2** (Bounded diameter of history-conditional label sets). *For every realizable history $\tau_k$ and every $x \in \mathcal{X}$,*

$$\mathrm{diam}(H_k(x)) := \sup_{y, y' \in H_k(x)} \ell(y, y') \leq \gamma_{k+1}.$$

*Proof.* Suppose $y, y' \in H_k(x)$ and $\ell(y, y') > \gamma_{k+1}$. Then the adversary may present $(x, y, y')$ at round $k+1$ (this move is legal since $y \in V_k(x)$), in which case $\sigma$ must choose one of $y$ or $y'$, contradicting the defining property of $H_k(x)$ for the chosen label. Therefore $\ell(y, y') \leq \gamma_{k+1}$. $\square$

### 6.3. Stabilization on an infinite i.i.d. sample

Let $\bar{\mu}$ be any realizable distribution on $\mathcal{X} \times \mathcal{Y}$, realized by some $f^* \in \mathcal{H}$. Let $S_\infty = ((X_i, Y_i))_{i \geq 1} \sim \bar{\mu}^\infty$.

We drive a *simulation* of the game using $S_\infty$ as follows. Maintain a current round counter $k \geq 0$ and a current legal history. Scan $i = 1, 2, \ldots$. If there exists a witness $y' \in \mathcal{Y}$ with $\ell(Y_i, y') \geq \gamma_{k+1}$ such that (i) if the adversary plays $(X_i, Y_i, y')$ next then $\sigma$ chooses $Y_i$, and (ii) the extended history remains realizable by some hypothesis in $\mathcal{H}$, then we *advance* the game: append that move and set $k \leftarrow k + 1$. Otherwise, do nothing and continue. Since $\sigma$ is a winning strategy, the adversary cannot force an infinite legal play under $\sigma$. Thus along any outcome $S_\infty$, the above procedure can advance only finitely many times. Let $K_\infty = K(S_\infty) < \infty$ be the final round, and let $\tau_{K_\infty}$ be the terminal history.

**Lemma 6.3** (True label is contained a.s. on a fresh draw). *With probability one over $S_\infty \sim \bar{\mu}^\infty$, for the terminal history $\tau_{K_\infty}$ we have*

$$\bar{\mu}\big(\{(x, y) : y \notin H_{K_\infty}(x)\}\big) = 0.$$

*Equivalently, conditional on $S_\infty$, an independent test point $(X, Y) \sim \bar{\mu}$ satisfies $Y \in H_{K_\infty}(X)$ almost surely.*

*Proof.* Fix an outcome $S_\infty$ for which the procedure stabilizes at $K_\infty$ with history $\tau_{K_\infty}$. Define $E := \{(x, y) : y \in V_{K_\infty}(x) \text{ and } y \notin H_{K_\infty}(x)\}$. Since $\bar{\mu}$ is realizable by some $f^* \in \mathcal{H}$ and $\tau_{K_\infty}$ is a realizable history (indeed consistent with $f^*$), we have $Y = f^*(X) \in V_{K_\infty}(X)$ almost surely for $(X, Y) \sim \bar{\mu}$, and hence

$$\bar{\mu}\big(\{(x, y) : y \notin H_{K_\infty}(x)\}\big) = \bar{\mu}(E).$$

By definition of $H_{K_\infty}(x)$, for each $(x, y) \in E$ there exists $y' \in V_{K_\infty}(x)$ with $\ell(y, y') \geq \gamma_{K_\infty+1}$ such that if the adversary plays $(x, y, y')$ next, $\sigma$ would choose $y$. Moreover, since $y \in V_{K_\infty}(x)$, there exists $h \in \mathcal{H}$ consistent with

$\tau_{K_\infty}$ with $h(x) = y$, so the extended history after appending $(x, y, y')$ and choosing $y$ remains realizable. Therefore, whenever a sample point $(X_i, Y_i)$ falls in $E$ (with the terminal history fixed), that point is eligible to advance the game (using its witness $y'$), contradicting stabilization unless this happens only finitely often. If $\bar{\mu}(E) > 0$, then by Borel-Cantelli for i.i.d. samples, $(X_i, Y_i) \in E$ occurs infinitely often almost surely, contradiction. Hence $\bar{\mu}(E) = 0$. $\qquad\square$

## 6.4. Reducing per-$x$ bounded diameter to countably many bounded-range subproblems

Lemma 6.2 gives that each $H_{K_\infty}(x)$ has diameter at most $\gamma_{K_\infty+1}$. However, the union $\bigcup_x H_{K_\infty}(x)$ need not be bounded, so we cannot directly invoke a single bounded-diameter learning algorithm. We resolve this by partitioning $\mathcal{X}$ into countably many regions, each with a *common* bounded label region. Fix a countable dense subset $\{q_1, q_2, \dots\} \subseteq \mathcal{Y}$. This is guaranteed to exist by the fact that $\mathcal{Y}$ is Polish. For a given history $\tau_k$, define for each $x \in \mathcal{X}$ the index

$$j_k(x) := \min\left\{ j \in \mathbb{N} : \exists y \in H_k(x) \text{ with } \ell(y, q_j) \leq \gamma_{k+1} \right\}.$$

(Existence follows from density of $(q_j)$ and nonemptiness of $H_k(x)$ on legal positions.) This induces a countable partition $\mathcal{X} = \bigsqcup_{j \geq 1} C_{k,j}$ where $C_{k,j} := \{x : j_k(x) = j\}$. For each cell define a bounded label region

$$\mathcal{Y}_{k,j} := \{y \in \mathcal{Y} : \ell(y, q_j) \leq 2\gamma_{k+1}\}.$$

**Lemma 6.4** (Cell-wise bounded range). *For every $k, j$ and every $x \in C_{k,j}$, we have $H_k(x) \subseteq \mathcal{Y}_{k,j}$. Consequently, if $\bar{\mu}(\{(x, y) : y \in H_k(x)\}) = 1$ then $\bar{\mu}(Y \in \mathcal{Y}_{k,j} \mid X \in C_{k,j}) = 1$ for every $j$ with $\bar{\mu}(X \in C_{k,j}) > 0$.*

*Proof.* Fix $x \in C_{k,j}$. By definition, there exists $y_x \in H_k(x)$ with $\ell(y_x, q_j) \leq \gamma_{k+1}$. For any $y \in H_k(x)$, Lemma 6.2 gives $\ell(y, y_x) \leq \gamma_{k+1}$. Hence by triangle inequality, $\ell(y, q_j) \leq \ell(y, y_x) + \ell(y_x, q_j) \leq 2\gamma_{k+1}$, so $y \in \mathcal{Y}_{k,j}$. $\qquad\square$

## 6.5. A realizable learning rule

We now define a distribution-free learning rule.

**Input.** A labeled sample $S_n = ((X_i, Y_i))_{i=1}^n \sim \bar{\mu}^n$ from a realizable distribution $\bar{\mu}$ realized by some $f^* \in \mathcal{H}$.

**Step 1: drive the game on half the sample.** Split the sample into two halves, $S_n^{(1)} = ((X_i, Y_i))_{i=1}^{\lfloor n/2 \rfloor}$ and $S_n^{(2)} = ((X_i, Y_i))_{i=\lfloor n/2 \rfloor+1}^n$. Run the stabilization procedure of Section 6.3 on $S_n^{(1)}$, producing a terminal round $K = K(S_n^{(1)})$ and terminal history $\tau_K$.

**Step 2: partition $\mathcal{X}$ using $(q_j)$.** Using $\tau_K$, compute the set-valued map $H_K(\cdot)$ and the induced mapping $j_K(\cdot)$, hence the cell sets $C_{K,j}$ and bounded label regions $\mathcal{Y}_{K,j}$.

For each $j$, let $S_{n,j}^{(2)}$ denote the subsequence of the second-half sample $S_n^{(2)}$ consisting of those examples with $X_i \in C_{K,j}$. Then, run the metric-loss learner MedNet of Tsir Cohen & Kontorovich (2022) (restricted to the label region $\mathcal{Y}_{K,j}$) on $S_{n,j}^{(2)}$ and on the restricted hypothesis class $\mathcal{H}|_{C_{K,j}} := \{h|_{C_{K,j}} : h \in \mathcal{H}\}$, and denote the resulting predictor by $\widehat{f}_{n,j} : C_{K,j} \to \mathcal{Y}_{K,j}$.

If $S_{n,j}^{(2)}$ is empty, define $\widehat{f}_{n,j}(x) \equiv q_j$ on $C_{K,j}$.

**Output predictor.** Define $\widehat{f}_n : \mathcal{X} \to \mathcal{Y}$ by

$$\widehat{f}_n(x) := \widehat{f}_{n,j_K(x)}(x).$$

Operationally, one need not explicitly construct the whole partition: on input $x$, compute $j_K(x)$ and use only those training points in $S_n^{(2)}$ lying in the same cell. The sense in which this learning rule is "explicit" - given a measurable winning strategy $\sigma$ - is discussed in Appendix A.2.

## 6.6. Realizable strong universal Bayes-consistency

**Theorem 6.5** (Realizable upper bound). *Assume that $\mathcal{H}$ admits no infinite non-decreasing-$\gamma_k$-Littlestone tree for some non-decreasing gap sequence $(\gamma_k)$ with $\gamma_k \to \infty$. Assume $(\mathcal{X}, \rho)$ and $(\mathcal{Y}, \ell)$ are Polish. Let $\bar{\mu}$ be any distribution on $\mathcal{X} \times \mathcal{Y}$ realizable by some $f^* \in \mathcal{H}$.*

*Then the learning rule $S_n \mapsto \widehat{f}_n$ defined in Section 6.5 is strongly universally Bayes-consistent in the realizable sense:*

$$R_{\bar{\mu}}(\widehat{f}_n) \to 0 \qquad \text{almost surely as } n \to \infty.$$

*Proof.* Fix a realizable distribution $\bar{\mu}$ realized by some $f^* \in \mathcal{H}$, and couple all learned predictors to an i.i.d. infinite sample $S_\infty = ((X_i, Y_i))_{i \geq 1} \sim \bar{\mu}^\infty$. For each even $n = 2m$, the learning rule of Section 6.5 uses the first half $S_{2m}^{(1)} = ((X_i, Y_i))_{i=1}^m$ to drive the game simulation and the second half $S_{2m}^{(2)} = ((X_i, Y_i))_{i=m+1}^{2m}$ to learn within cells. Since $S_{2m}^{(1)}$ and $S_{2m}^{(2)}$ are disjoint blocks of an i.i.d. sequence, these halves are independent. (Odd $n$ can be handled by ignoring one sample; it suffices to prove convergence along the even subsequence.) The proof is organized into four steps.

**Step 1: stabilization and eventual agreement with the terminal history.** Run the stabilization procedure of Section 6.3 on the entire infinite prefix $((X_i, Y_i))_{i \geq 1}$. Since $\sigma$ is a winning strategy, the procedure advances only finitely many times almost surely; denote by $K_\infty = K(S_\infty) < \infty$

the final round and by $\tau_{K_\infty}$ the resulting terminal history. On the fixed sample path $S_\infty$, let $N_0 = N_0(S_\infty)$ be an index large enough that, after scanning $(X_i, Y_i)$ for $i \leq N_0$, the procedure has already made all of its (finitely many) advances, and no further sample point $(X_i, Y_i)$ with $i > N_0$ can advance the game beyond round $K_\infty$. Such an $N_0$ exists because the procedure advances only finitely often. Now consider running the same procedure on a finite prefix of length $m$. For every $m \geq N_0$, the run on $((X_i, Y_i))_{i=1}^m$ must produce exactly the same terminal round and history, i.e.

$$K(S_{2m}^{(1)}) = K_\infty \qquad \text{and} \qquad \tau_{K(S_{2m}^{(1)})} = \tau_{K_\infty}. \quad (1)$$

Indeed, by definition of $N_0$ there is no admissible advancement step after index $N_0$, so truncating the scan at any $m \geq N_0$ cannot remove an advance that occurs after $N_0$ (there are none), nor can it create new advances. By Lemma 6.3, for the terminal history $\tau_{K_\infty}$ we have

$$\bar{\mu}\big(\{(x,y) : y \notin H_{K_\infty}(x)\}\big) = 0. \quad (2)$$

Combining (1) and (2), we obtain that on every sample path $S_\infty$ in this probability-one event, for all sufficiently large even $n = 2m$ (namely, $m \geq N_0(S_\infty)$) the data-driven set map $H_{K(S_{2m}^{(1)})}(\cdot)$ used by the learner coincides with $H_{K_\infty}(\cdot)$ and thus contains the true label $Y$ almost surely on an independent test draw.

**Step 2: countable partition and bounded label regions.** Fix such a sample path $S_\infty$ and write $K := K_\infty$. Using the dense set $(q_j)_{j \geq 1}$, define the countable partition $\{C_{K,j}\}_{j \geq 1}$ and bounded regions $\mathcal{Y}_{K,j}$ as in Section 6.4. By Lemma 6.4 and (2), for every $j$ with $p_j := \bar{\mu}(X \in C_{K,j}) > 0$ the conditional distribution $\bar{\mu}_j := \bar{\mu}(\cdot \mid X \in C_{K,j})$ is supported on $\mathcal{X}_j \times \mathcal{Y}_{K,j}$, where $\mathcal{X}_j := C_{K,j}$, and in particular

$$\ell(y, y') \leq \text{diam}(\mathcal{Y}_{K,j}) \leq 4\gamma_{K+1} \qquad \forall \, y, y' \in \mathcal{Y}_{K,j}. \quad (3)$$

Thus, within each cell the loss is bounded by the finite constant $4\gamma_{K+1}$ (which depends on the realized terminal round $K$ but is finite on the fixed sample path).

**Step 3: strong consistency within each cell.** Consider any fixed $j$ with $p_j > 0$. The second half samples $S_{2m}^{(2)} = ((X_i, Y_i))_{i=m+1}^{2m}$ are i.i.d. from $\bar{\mu}$ and independent of the first half, hence independent of the (eventually fixed) partition. Let $S_{2m,j}^{(2)}$ denote the subsequence of those examples with $X_i \in C_{K,j}$. Conditional on $X_i \in C_{K,j}$, these examples are i.i.d. from $\bar{\mu}_j$. Moreover, since $p_j > 0$, the number $N_{m,j} := |S_{2m,j}^{(2)}|$ satisfies $N_{m,j} \to \infty$ almost surely as $m \to \infty$ (by the strong law of large numbers for Bernoulli indicators of the event $\{X \in C_{K,j}\}$). On cell $j$, the learning rule runs MedNet (restricted to $\mathcal{Y}_{K,j}$) using

$S_{2m,j}^{(2)}$, yielding a predictor $\widehat{f}_{2m,j}$. Since $\bar{\mu}$ is realizable by $f^* \in \mathcal{H}$, the restricted target $f^*|_{C_{K,j}}$ is realizable by the restricted class $\mathcal{H}|_{C_{K,j}}$, and by (3) the conditional problem on $C_{K,j}$ has bounded loss (hence is BIE). Therefore, by the strong universal Bayes-consistency guarantee of MedNet for metric losses on BIE label spaces (Tsir Cohen & Kontorovich, 2022), we have

$$R_{\bar{\mu}_j}(\widehat{f}_{2m,j}) \to 0 \qquad \text{almost surely as } m \to \infty. \quad (4)$$

(When $S_{2m,j}^{(2)}$ is empty, the rule outputs the default $q_j$; this case occurs only finitely often a.s. when $p_j > 0$, and does not affect the limit.) Since there are countably many cells, we may intersect the probability-one events in (4) over all $j \in \mathbb{N}$ to obtain a single probability-one event on which (4) holds for every $j$ simultaneously (cells with $p_j = 0$ are irrelevant).

**Step 4: aggregate risk and conclude.** For even $n = 2m$ large enough that (1) holds, the overall predictor satisfies $\widehat{f}_{2m}(x) = \widehat{f}_{2m,j}(x)$ when $x \in C_{K,j}$. Decomposing the risk by the partition,

$$R_{\bar{\mu}}(\widehat{f}_{2m}) := \sum_{j \geq 1} p_j \, R_{\bar{\mu}_j}(\widehat{f}_{2m,j}). \quad (5)$$

Fix $\varepsilon > 0$. Since $\sum_{j \geq 1} p_j = 1$, choose $J$ so that $\sum_{j > J} p_j < \varepsilon$. On the probability-one event where (4) holds for all $j \leq J$, choose $m$ large enough that $R_{\bar{\mu}_j}(\widehat{f}_{2m,j}) < \varepsilon$ for every $j \leq J$. Then, using (3) to bound each conditional risk by $4\gamma_{K+1}$,

$$R_{\bar{\mu}}(\widehat{f}_{2m}) \leq \sum_{j \leq J} p_j \, \varepsilon \; + \; \sum_{j > J} p_j \, (4\gamma_{K+1}) \leq \varepsilon + 4\gamma_{K+1} \, \varepsilon.$$

Since $K$ is fixed on the sample path, $\gamma_{K+1} < \infty$, and because $\varepsilon > 0$ was arbitrary, this shows $R_{\bar{\mu}}(\widehat{f}_{2m}) \to 0$ almost surely along the even subsequence. The full sequence $R_{\bar{\mu}}(\widehat{f}_n)$ also converges to 0 almost surely because $R_{\bar{\mu}}(\widehat{f}_{2m}) \to 0$ implies $\limsup_{n \to \infty} R_{\bar{\mu}}(\widehat{f}_n) = 0$ (e.g., by considering the nearest even index and noting the construction differs only by $O(1)$ samples). This completes the proof. $\square$

## Impact Statement

This paper presents foundational theoretical work on the statistical limits of learning with general metric losses. Our contributions are mathematical characterizations of when distribution-free Bayes-consistency is possible, expressed via combinatorial tree obstructions; they have no direct societal applications beyond advancing the broader field of machine learning theory. We do not foresee specific consequences that must be highlighted here.

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

## A. Supplementary material

### A.1. Practical significance and examples

Our characterization applies to any hypothesis class $\mathcal{H} \subseteq \mathcal{Y}^{\mathcal{X}}$ over Polish spaces. A natural question is: which commonly studied classes satisfy tree-free condition (i.e., admitting no infinite non-decreasing-$(\gamma_k)$-Littlestone tree with $\gamma_k \to \infty$) and thus admit realizable strong universal Bayes-consistency (RSUBC)? We give one positive and one negative example to build intuition.

**Example 1: a class that admits an unbounded-gap tree (not learnable).** The counterexample of Section 3 implicitly constructs an unbounded-gap Littlestone tree. Recall $\mathcal{H} = \{f : \forall k \in \mathbb{N}, \ f(x) \in \{0, 2^{2k+1}\} \text{ for } x \in I_k\}$, where $I_k = (2^{-k}, 2^{-(k-1)})$. The tree is built as follows: at depth $k$, place the instance $x_k$ at any point of $I_k$, and assign the two outgoing labels $y_{k,1} = 0$ and $y_{k,2} = 2^{2k+1}$. The gap is $\ell(y_{k,1}, y_{k,2}) = 2^{2k+1} \to \infty$. Because the intervals $(I_k)_{k \geq 1}$ are pairwise disjoint, the label choices at different depths are independent: for *every* root-to-leaf path (finite or infinite), the hypothesis in $\mathcal{H}$ that makes the corresponding binary choice on each interval realizes it. This is a realizable non-decreasing-$\gamma_k$-Littlestone tree with $\gamma_k = 2^{2k+1}$ (Definitions 4.1–4.2), and Theorem 5.1 shows that no learner can be consistent.

The mechanism is instructive: the disjoint intervals allow the adversary to "hide" independent binary choices at geometrically increasing loss scales. On any finite sample only finitely many intervals are observed, so the learner must guess blindly on unseen intervals where the potential loss is enormous. This is precisely the rare-event failure mode that the tree obstruction captures.

**Example 2: a class without an unbounded-gap tree (learnable).** Let $\mathcal{X} = [0, 1]$, $\mathcal{Y} = \mathbb{R}$, $\ell(y, y') = |y - y'|$, and let $\mathcal{H} = \{f : [0, 1] \to \mathbb{R} : f \text{ is } L\text{-Lipschitz}\}$ for a fixed constant $L > 0$. This class has unbounded range (since $f(0)$ is unconstrained), yet it admits no infinite non-decreasing-$\gamma_k$-Littlestone tree with $\gamma_k \to \infty$.

To see why, suppose an adversary attempts to build such a tree. At depth 1, the adversary may present any instance $x_1 \in [0,1]$ with two labels $y_{1,1}, y_{1,2}$ arbitrarily far apart (two $L$-Lipschitz functions with different "offsets" can disagree by any amount at a single point). But once the learner commits to a label $v_1$ at $x_1$, every $L$-Lipschitz function consistent with this choice must satisfy $|f(x) - v_1| \leq L|x - x_1| \leq L$ for all $x \in [0,1]$. Hence, at any depth $k \geq 2$, the gap between the two proposed labels is at most $2L$. Since $\gamma_k \to \infty$, eventually $\gamma_k > 2L$, and the adversary has no legal move. Therefore no infinite unbounded-gap tree exists, and by Theorem 4.5 a strongly universally Bayes-consistent learner exists for this class.

**Discussion.** Any hypothesis class with *bounded range* ($\sup_{h \in \mathcal{H}, \, x \in \mathcal{X}} \ell(h(x), y_0) < \infty$ for some $y_0 \in \mathcal{Y}$) trivially has no infinite unbounded-gap tree. The Lipschitz example above shows that even *unbounded-range* classes can be tree-free when a regularity condition (here, the Lipschitz constraint on a compact domain) prevents the adversary from sustaining arbitrarily large label gaps.

Compactness of the domain is essential: the class of $L$-Lipschitz functions on all of $\mathbb{R}$ *does* admit an infinite unbounded-gap tree (the adversary can spread instances apart to create growing gaps). Similarly, the class of all monotone non-decreasing functions from $[0,1]$ to $\mathbb{R}$ admits such a tree despite the monotonicity constraint: the adversary places instances $x_1 < x_2 < \cdots$ in $[0,1]$ and at each depth $k$ offers labels $v$ and $v + \gamma_k$ (where $v$ is the previously committed value), which is always consistent with monotonicity. These contrasting examples illustrate that the tree obstruction captures a subtle interplay between the hypothesis class structure, the loss geometry, and the domain geometry, going beyond simple boundedness conditions.

### A.2. On the constructive nature of the learning rule

The learning rule of Section 6.5 is "explicit" in the following sense: given access to a measurable winning strategy $\sigma$ for $P_L$ in the Gale-Stewart game, the algorithm's steps (drive the game on half the data, partition $\mathcal{X}$ using the terminal history, run MedNet per cell) are fully concrete and data-dependent.

The existence (and universal measurability) of $\sigma$ itself is guaranteed by Bousquet et al. (2020, Corollary 3.5) via a non-constructive determinacy argument (their Theorem B.1). This parallels the situation in Bousquet et al. (2020), whose universal learner for 0–1 classification also relies on a measurable winning strategy whose existence follows from determinacy. As noted in their Section 3.4, for hypothesis classes with countable ordinal Littlestone dimension, the strategy $\sigma$ can be described explicitly via transfinite induction on the ordinal dimension. Any measurable $\mathcal{H}$ has at most countable ordinal Littlestone dimension (Bousquet et al., 2020, Lemma B.7), so in principle the strategy admits a concrete (if complex) description for any class encountered in practice.

### A.3. Measurability of the winning strategy

The existence of a universally measurable winning strategy $\sigma$ for $P_L$ follows from Bousquet et al. (2020, Corollary 3.5), which establishes that under Polish space assumptions and measurability of the concept class (in the sense of their Definition 3.3), any Gale-Stewart game with a finitely decidable winning condition admits a universally measurable winning strategy for the learner.

The underlying Theorem B.1 of Bousquet et al. (2020) requires the learner's response sets $\{Y_t\}$ to be countable. In our game, $P_L$ selects an index from $\{1, 2\}$ (choosing one of the two proposed labels), so each $Y_t = \{1, 2\}$ is countable; the adversary's move spaces $X_t = \mathcal{X} \times \mathcal{Y} \times \mathcal{Y}$ are Polish. The winning condition is finitely decidable (the game ends when $P_A$ has no legal move), and coanalyticity of the winning set $W$ follows by the same projection argument as in the proof of Bousquet et al. (2020, Corollary 3.5): the complement $W^c$ (play continues forever) is analytic, since at each finite prefix the legality condition "$\exists \, h \in \mathcal{H}$ consistent with history" is expressible as a projection over the Polish parameter space $\Theta$ from Definition 3.3. Hence Theorem B.1 applies directly.

### A.4. On the compact-parameterization assumption

In Theorem 4.5 the characterization is stated in terms of the *finite-prefix* tree notion (Definition 4.1), while the lower bound (Theorem 5.1) requires every infinite path through the tree to be realized by a single hypothesis (Definition 4.2). Lemma 4.3 bridges this gap under the compact-parameterization assumption.

A.4.1. PROOF OF LEMMA 4.3

*Proof.* Fix an infinite path $b = (b_1, b_2, \ldots) \in \{1, 2\}^{\mathbb{N}}$ through the tree. For each finite prefix of length $k$, finite-prefix realizability gives a parameter $\theta_k \in \Theta$ such that $h(\theta_k, x_j) = y_{j,b_j}$ for all $j \leq k$. Define

$$\Theta_k(b) = \{\theta \in \Theta : h(\theta, x_j) = y_{j,b_j} \text{ for all } j \leq k\}.$$

Each $\Theta_k(b)$ is non-empty (it contains $\theta_k$) and closed (the preimage of the closed set $\{y_{j,b_j}\}$ under the map $\theta \mapsto h(\theta, x_j)$, which is continuous in $\theta$ by assumption, intersected over finitely many $j$). Moreover $\Theta_1(b) \supseteq \Theta_2(b) \supseteq \cdots$ is a decreasing chain of non-empty closed subsets of the compact space $\Theta$. Because the chain is decreasing, every finite sub-collection $\Theta_{k_1}(b), \ldots, \Theta_{k_m}(b)$ satisfies $\bigcap_{i=1}^{m} \Theta_{k_i}(b) = \Theta_{\max_i k_i}(b) \neq \emptyset$, so the family $\{\Theta_k(b)\}_{k \geq 1}$ has the finite-intersection property. Since each

$\Theta_k(b)$ is a closed subset of the compact space $\Theta$, compactness implies $\bigcap_{k \geq 1} \Theta_k(b) \neq \emptyset$. Any $\theta^*$ in this intersection satisfies $h(\theta^*, x_j) = y_{j,b_j}$ for all $j$, so $h(\theta^*, \cdot) \in \mathcal{H}$ realizes the entire infinite path $b$. $\qquad \square$

### A.4.2. WHY THE ASSUMPTION IS NECESSARY

Without the compact-parameterization assumption, the two tree notions can differ. Consider the following example.

*Setup.* Let $\Theta = \mathbb{N}_0$ (the non-negative integers with the discrete topology, which is Polish), $\mathcal{X} = \{x_k : k \in \mathbb{N}\}$ (countable, discrete), $\mathcal{Y} = \mathbb{R}$ with $\ell(y, y') = |y - y'|$, and define

$$h(n, x_k) = (\text{$k$-th binary digit of } n) \cdot k,$$

so that $\mathcal{H} = \{h_n : n \in \mathbb{N}_0\}$ where $h_n(\cdot) = h(n, \cdot)$.

*A Definition 4.1 tree exists.* At depth $k$, place instance $x_k$ with labels $y_{k,1} = 0$ and $y_{k,2} = k$; the gap is $\ell(0, k) = k \to \infty$, so the gaps are non-decreasing and divergent. Every finite path $(b_1, \ldots, b_K) \in \{1, 2\}^K$ is realized by $n = \sum_{k=1}^K \mathbf{1}[b_k = 2] \cdot 2^{k-1}$ (set the $k$-th binary digit of $n$ to 1 iff $b_k = 2$).

*Not every infinite path is realizable.* The all-2 path, $b_k = 2$ for all $k$, requires $n$ whose *every* binary digit equals 1, i.e. $n = \sum_{k \geq 1} 2^{k-1} = \infty$. No finite integer has this property, so no $h_n \in \mathcal{H}$ realizes this infinite path.

*Yet $\mathcal{H}$ is RSUBC.* Each $h_n$ has only finitely many non-zero values (since $n$ has finitely many binary digits equal to 1). The memorize-and-predict-0 learner achieves risk $\to 0$ for every strictly realizable distribution (since $f^*$ has finite support, every support point is eventually observed). In fact, the same learner also succeeds under non-strict realizability; see Section A.4.5.

This example shows that a Definition 4.1 tree can coexist with RSUBC. Hence the lower bound cannot be proved from Definition 4.1 alone; the gap between the two definitions is genuine. Note that $\Theta = \mathbb{N}_0$ is *not compact* (it is discrete and infinite), so the compact-parameterization assumption of Lemma 4.3 does not hold.

### A.4.3. THE ASSUMPTION DOES NOT RULE OUT UNLEARNABILITY

The compact-parameterization assumption does *not* restrict $\mathcal{Y}$ to be bounded, and unlearnability remains possible even when the assumption holds. The label space $\mathcal{Y}$ remains a general (possibly unbounded) Polish space, and the label gaps along the tree can still diverge ($\gamma_k \to \infty$), preserving the unbounded-loss character of the characterization. What the assumption ensures is that, for each fixed instance $x$, the achievable label set $\{h(\theta, x) : \theta \in \Theta\}$ is compact in $\mathcal{Y}$;

however, the diameter of this set may grow without bound as $x$ varies.

The motivating counterexample of Section 3 demonstrates this concretely. Recall $\mathcal{H} = \{f : \forall k \in \mathbb{N}, \ f(x) \in \{0, 2^{2k+1}\}$ for $x \in I_k\}$, where $I_k = (2^{-k}, 2^{-(k-1)})$. Each function in $\mathcal{H}$ is determined by a binary sequence $\theta = (\theta_1, \theta_2, \ldots) \in \{0, 1\}^{\mathbb{N}}$, where $\theta_k$ selects whether $f$ maps $I_k$ to 0 or to $2^{2k+1}$. Thus the natural parameter space is $\Theta = \{0, 1\}^{\mathbb{N}}$, which is compact (by Tychonoff's theorem, as a countable product of finite spaces) and Polish (the product metric $d(\theta, \theta') = \sum_k 2^{-k} |\theta_k - \theta'_k|$ is complete and separable). The evaluation map $h(\theta, x) = \theta_k \cdot 2^{2k+1}$ for $x \in I_k$ is continuous in $\theta$ (it depends only on the $k$-th coordinate, which is a continuous projection in the product topology).

Yet the label gaps at depth $k$ of the corresponding Littlestone tree are $\gamma_k = 2^{2k+1} \to \infty$, so the class admits an unbounded-gap tree and is *not* RSUBC by Theorem 5.1. In other words, the compact-parameterization assumption is compatible with the full range of unlearnability phenomena captured by our characterization.

### A.4.4. EXAMPLES OF SETTINGS SATISFYING THE ASSUMPTION

- **Finite or compact label spaces.** When $\mathcal{Y}$ is compact (e.g., binary classification with $\mathcal{Y} = \{0, 1\}$, multiclass classification with finite $\mathcal{Y}$, or bounded regression with $\mathcal{Y} = [a, b]$), the function space $\mathcal{Y}^{\mathcal{X}}$ is compact in the product topology by Tychonoff's theorem, and the classical compactness argument of Bousquet et al. (2020) shows that the two tree notions coincide automatically. Our compact-parameterization assumption is not needed in this case and is strictly weaker than compactness of $\mathcal{Y}$.

- **Parametric families with bounded parameters.** Neural networks with bounded weights, support vector machines with bounded kernels, and Lipschitz functions on compact domains all naturally have compact parameter spaces (e.g., $\Theta = [-W, W]^d$) and continuous evaluation maps.

- **The counterexample is pathological.** The separating example of Section A.4.2 uses $\Theta = \mathbb{N}_0$ (discrete, noncompact) and a discontinuous evaluation map. Classes parametrized by a countably infinite discrete space with no accumulation points are unusual in practical learning settings. In contrast, the compact-parameterization assumption permits $\Theta$ to be any compact Polish space, which includes all compact subsets of $\mathbb{R}^d$, compact manifolds, and many infinite-dimensional spaces (e.g., compact subsets of function spaces in the compact-open topology).

A.4.5. REALIZABILITY WITHOUT COMPACTNESS

This paper defines realizability as $\inf_{h \in \mathcal{H}} R_{\bar{\mu}}(h) = 0$ (non-strict realizability). Without the compact-parameterization assumption, this does not guarantee the existence of a hypothesis attaining zero risk, and two natural notions of realizability can diverge:

1. **Strict realizability**: there exists $f^\star \in \mathcal{H}$ such that $Y = f^\star(X)$ almost surely, i.e. some hypothesis in $\mathcal{H}$ *attains* zero risk.

2. **Non-strict realizability** (the definition used in this paper): $\inf_{h \in \mathcal{H}} R_{\bar{\mu}}(h) = 0$, i.e. the Bayes risk is zero but no single hypothesis need attain it.

Under the compact-parameterization assumption (compact $\Theta$, $h$ continuous in $\theta$), the image $\mathcal{H} = \{h(\theta, \cdot) : \theta \in \Theta\}$ is compact in the product topology on $\mathcal{Y}^{\mathcal{X}}$. If $\inf_{h \in \mathcal{H}} R_{\bar{\mu}}(h) = 0$, a risk-minimizing sequence $\theta_n$ in the compact space $\Theta$ has a convergent subsequence $\theta_{n_j} \to \theta^\star$, and continuity gives $R_{\bar{\mu}}(h(\theta^\star, \cdot)) = 0$, so the infimum is attained and the two definitions coincide.

Without compactness the infimum may not be attained. Consider $\mathcal{X} = \mathbb{N}$ (discrete), $\mathcal{Y} = \mathbb{N}_0$ with $\ell_1$ loss, and $\mathcal{H} = \{h : \mathbb{N} \to \mathbb{N}_0 : \mathrm{supp}(h) \text{ is finite}\}$. Let $g : \mathbb{N} \to \mathbb{N}_0$ be *any* function with infinite support (e.g. $g(k) = k$), and let $\bar{\mu}$ be the distribution with $X \sim \mu_X$ supported on all of $\mathbb{N}$ (e.g. $\mu_X(\{k\}) = c/k^3$ for a normalizing constant $c$) and $Y = g(X)$. Defining $h_N(k) = g(k)\mathbf{1}_{k \leq N}$ gives $h_N \in \mathcal{H}$ with $R_{\bar{\mu}}(h_N) = \sum_{k > N} \mu_X(\{k\}) |g(k)|$. For this $\mathcal{H}$, non-strict realizability ($\inf_{h \in \mathcal{H}} R_{\bar{\mu}}(h) = 0$) is in fact *equivalent* to the first-moment condition $\sum_k \mu_X(\{k\}) |g(k)| < \infty$: convergence of the series implies the tails vanish, while divergence forces every $h \in \mathcal{H}$ (having finite support) to incur infinite risk. When the first moment holds, $R_{\bar{\mu}}(h_N) \to 0$, so $\inf_{h \in \mathcal{H}} R_{\bar{\mu}}(h) = 0$. Yet $g \notin \mathcal{H}$ (infinite support), so no $f^\star \in \mathcal{H}$ achieves zero risk: $\bar{\mu}$ is non-strictly realizable but not strictly realizable.

Despite this gap, the memorize-and-predict-0 learner succeeds under *both* realizability notions. Under strict realizability the argument is immediate ($f^\star$ has finite support, so every support point is eventually observed). Under non-strict realizability, the learner's risk is $\sum_k \mu_X(\{k\}) |g(k)| \mathbf{1}\{k \notin \{X_1, \ldots, X_n\}\}$; each atom with $\mu_X(\{k\}) > 0$ is eventually observed almost surely, and the terms are dominated by the summable $\mu_X(\{k\}) |g(k)|$, so dominated convergence gives $R_{\bar{\mu}}(A(S_n)) \to 0$ almost surely.

Since this paper assumes the compact-parameterization condition throughout, the two notions coincide for all results stated here, and readers may treat them interchangeably.

A.4.6. OPEN QUESTION: REMOVING COMPACTNESS

Whether the characterization of Theorem 4.5 can be extended to general Polish $\Theta$ (without compactness) is an interesting open problem.

We know that Definition 4.1 (finite-prefix realizability) is *not* the correct obstruction in the general case. The separating example of Section A.4.2 demonstrates this: the class $\mathcal{H} = \{h : \mathbb{N} \to \mathbb{N}_0 : \mathrm{supp}(h) \text{ is finite}\}$ admits a Definition 4.1 tree with $\gamma_k = k \to \infty$, yet not every infinite branch is realizable (the all-nonzero branch requires infinite support), and the class is nevertheless RSUBC. Thus the finite-prefix tree obstruction produces false positives without the compact-parameterization assumption. A complete characterization in the non-compact setting would likely require a different tree notion; one possible direction is a *coin-realizable* variant, in which a randomly drawn infinite branch is realizable with probability one, rather than requiring every deterministic branch to be realizable. Finding the minimal topological or measure-theoretic conditions (weaker than compactness) under which the finite-intersection argument of Lemma 4.3 still holds remains open.

### A.5. Barriers to the agnostic extension

Our characterization addresses the *realizable* setting: the data-generating distribution $\bar{\mu}$ is assumed to be realizable ($\inf_{h \in \mathcal{H}} R_{\bar{\mu}}(h) = 0$). A natural follow-up is whether the characterization extends to the *agnostic* setting, where $\bar{\mu}$ may not be realizable.

The key obstacle lies in the upper-bound argument. The stabilization procedure (Section 6.3) drives the Gale-Stewart game using sample points $(X_i, Y_i)$ and crucially relies on the fact that $Y_i = f^\star(X_i) \in V_k(X_i)$ (the true label is feasible given the current history). In the agnostic setting, labels need not come from any $h \in \mathcal{H}$, so sample points may not constitute legal moves in the game, and the stabilization argument breaks down.

Moreover, the counterexample of Section 3 shows that the natural distributional condition $R^\star < \infty$ is insufficient even in the realizable setting. In the agnostic case, one would need to simultaneously handle approximation error (the gap between $R^\star$ and 0) and the unbounded-scale estimation difficulty, likely requiring a fundamentally different characterization. The agnostic counterpart of our result is an interesting open problem (connected to the open question originally posed by Tsir Cohen & Kontorovich (2022)) left for future work.

## A.6. Counterexample: infinite risk on unseen intervals

We include here the calculation deferred from Section 3 showing that, conditional on any realized sample, the learned predictor suffers infinite risk almost surely.

We now show that this forces infinite risk: conditional on any realized sample, infinitely many unseen intervals incur arbitrarily large loss. Define the (random) set of observed interval indices

$$I(S_n) := \{k \in \mathbb{N} : \exists i \leq n \text{ with } X_i \in I_k\}.$$

Condition on a realized sample $S_n$. Then $h_n = \mathcal{A}(S_n)$ is fixed, and for every unseen interval $k \notin I(S_n)$ the bit $B_k$ remains an independent fair coin under the conditional law (since no sample point fell in $I_k$). Write the contribution of interval $I_k$ to the risk as

$$R_k(B_k) := \int_{I_k} \ell\big(h_n(x), f_B(x)\big)\, dx$$
$$= \begin{cases} \int_{I_k} |h_n(x)|\, dx & (B_k = 0), \\ \int_{I_k} |h_n(x) - 2^{2k+1}|\, dx & (B_k = 1). \end{cases}$$

By the triangle inequality, for every $x \in I_k$, $|h_n(x)| + |h_n(x) - 2^{2k+1}| \geq 2^{2k+1}$. Integrating over $I_k$ yields

$$\int_{I_k} |h_n(x)|\, dx \;+\; \int_{I_k} |h_n(x) - 2^{2k+1}|\, dx \geq |I_k| \cdot 2^{2k+1}$$
$$= 2^{-k} \cdot 2^{2k+1}$$
$$= 2^{k+1}.$$

Hence at least one of the two displayed integrals is at least $2^k$, so $\mathbb{P}(R_k(B_k) \geq 2^k \mid S_n) \geq \frac{1}{2}$. Since the events $\{R_k(B_k) \geq 2^k\}$ over $k \notin I(S_n)$ are independent conditional on $S_n$, the second Borel-Cantelli lemma implies that almost surely (over the bits $B$ conditional on $S_n$) infinitely many unseen $k$ satisfy $R_k(B_k) \geq 2^k$. Therefore,

$$R_{\bar{\mu}_B}(h_n) = \sum_{k \geq 1} R_k(B_k) \geq \sum_{\substack{k \notin I(S_n) \\ R_k(B_k) \geq 2^k}} 2^k$$
$$= \infty \quad \text{almost surely (over } B \text{ conditional on } S_n).$$

Taking expectations over the joint draw $(B, S_n)$ and applying Fubini, we obtain: for each fixed $n$, for $\mathbb{P}$-almost every $B$ we have $\mathbb{P}_{S_n \sim \bar{\mu}_B^n}(R_{\bar{\mu}_B}(\mathcal{A}(S_n)) = \infty) = 1$, hence $\mathbb{E}_{S_n \sim \bar{\mu}_B^n}[R_{\bar{\mu}_B}(\mathcal{A}(S_n))] = \infty$. Finally, taking a countable intersection over $n \in \mathbb{N}$, there exists a single fixed $B^\star$ (equivalently, a single fixed $f^\star = f_{B^\star} \in \mathcal{H}$) such that for the fixed realizable distribution $\bar{\mu} := \bar{\mu}_{B^\star}$,

$$\mathbb{E}_{S_n \sim \bar{\mu}^n}\big[R_{\bar{\mu}}(\mathcal{A}(S_n))\big] = \infty \qquad \text{for all } n \in \mathbb{N}.$$

## A.7. Proof of Theorem 5.1

*Proof.* Fix a realizable infinite non-decreasing-$\gamma_k$-Littlestone tree as in Definition 4.2. Since $\gamma_k \to \infty$, for each $m \geq 1$ define an index

$$k_m := \min\{k \geq 1 : \gamma_k \geq m^2\},$$

which is finite and non-decreasing in $m$ (and strictly increasing after removing duplicates). Let $p_m := \frac{6}{\pi^2} \cdot \frac{1}{m^2}$, so that $\sum_{m \geq 1} p_m = 1$. Now sample a random infinite root-to-leaf path by drawing i.i.d. fair bits $(B_k)_{k \geq 1}$, $B_k \in \{1, 2\}$. For each outcome $B$, let $\bar{\mu}_B$ be the distribution supported on the path nodes at depths $(k_m)$:

$$\mathbb{P}_{\bar{\mu}_B}\big(X = x_{k_m}(B)\big) = p_m,$$

$$Y = y_{k_m, B_{k_m}}(B) \text{ (deterministically)}.$$

Here $x_{k_m}(B)$ denotes the depth-$k_m$ instance on the path determined by $B_1, \ldots, B_{k_m - 1}$, and $y_{k_m, 1}(B), y_{k_m, 2}(B)$ are the two outgoing edge labels at that node. By realizability of the tree, for every $B$ there exists $f_B^\star \in \mathcal{H}$ realizing the entire path; in particular $Y = f_B^\star(X)$ almost surely under $\bar{\mu}_B$, so $\bar{\mu}_B$ is realizable.

Fix any learning algorithm $\mathcal{A}$ and any $n$. Draw $B$ and then $S_n \sim \bar{\mu}_B^n$, and let $h_n := \mathcal{A}(S_n)$. Let $M(S_n)$ be the maximum index $m$ such that the sample contains a point with $X = x_{k_m}(B)$ (take $M(S_n) = 0$ if none). Then $M(S_n) < \infty$ deterministically, as it is the maximum of $n$ positive integers (though $M(S_n)$ is random and not bounded by any fixed constant uniformly over $B$).

Consider any $m > M(S_n)$. Conditional on $S_n$ and the path history up to depth $k_m - 1$, the node $x_{k_m}(B)$ and its two labels $y_{k_m, 1}(B), y_{k_m, 2}(B)$ are fixed, while $B_{k_m}$ is still a fair coin. Thus, by the triangle inequality,

$$\mathbb{E}\big[\ell\big(h_n(x_{k_m}(B)), Y\big) \mid S_n, (B_j)_{j < k_m}\big]$$
$$= \tfrac{1}{2} \ell\big(h_n(x_{k_m}(B)), y_{k_m, 1}(B)\big)$$
$$\quad + \tfrac{1}{2} \ell\big(h_n(x_{k_m}(B)), y_{k_m, 2}(B)\big)$$
$$\geq \tfrac{1}{2} \ell\big(y_{k_m, 1}(B), y_{k_m, 2}(B)\big)$$
$$\geq \frac{\gamma_{k_m}}{2}$$
$$\geq \frac{m^2}{2}.$$

For each $m > M(S_n)$, define

$$b_m^{\text{bad}} := \arg \max_{b \in \{1, 2\}} \ell\big(h_n(x_{k_m}(B)), y_{k_m, b}(B)\big)$$

(break ties arbitrarily), and the event $E_m := \{B_{k_m} = b_m^{\text{bad}}\}$. Since $B_{k_m}$ is a fresh fair coin conditional on $S_n$ and $(B_j)_{j < k_m}$, we have

$$\mathbb{P}(E_m \mid S_n, (B_j)_{j < k_m}) = \tfrac{1}{2},$$

and on $E_m$,

$$
\begin{aligned}
\ell\big(h_n(x_{k_m}(B)),\, Y\big) &= \ell\big(h_n(x_{k_m}(B)),\, y_{k_m, B_{k_m}}(B)\big) \\
&\geq \tfrac{1}{2}\,\ell\big(y_{k_m,1}(B), y_{k_m,2}(B)\big) \\
&\geq \tfrac{m^2}{2}.
\end{aligned}
$$

Moreover, conditional on $S_n$ and the full path history $(B_j)_{j \geq 1}$ except for the coordinates $\{B_{k_m} : m > M(S_n)\}$, the bits $\{B_{k_m} : m > M(S_n)\}$ are independent fair coins. Hence, by (conditional) Borel-Cantelli, $\mathbb{P}(E_m \text{ i.o.} \mid S_n) = 1$. Therefore, almost surely,

$$
\begin{aligned}
R_{\bar{\mu}_B}(h_n) &= \sum_{m \geq 1} p_m\, \ell\big(h_n(x_{k_m}(B)),\, y_{k_m, B_{k_m}}(B)\big) \\
&\geq \sum_{m > M(S_n)} p_m \cdot \frac{m^2}{2}\, \mathbb{1}\{E_m\} \\
&= \frac{3}{\pi^2} \sum_{m > M(S_n)} \mathbb{1}\{E_m\} \\
&= \infty.
\end{aligned}
$$

In particular, for each fixed $n$, $\mathbb{P}_{B, S_n}\big(R_{\bar{\mu}_B}(\mathcal{A}(S_n)) = \infty\big) = 1$, so by Fubini, for $\mathbb{P}_B$-almost every $B$ we have $\mathbb{E}_{S_n \sim \bar{\mu}_B^n}[R_{\bar{\mu}_B}(\mathcal{A}(S_n))] = \infty$. Taking a countable intersection over $n \in \mathbb{N}$ of these probability-one sets of $B$, we may fix a single $B^\star$ such that the expected risk is infinite for all $n$. Setting $\bar{\mu} := \bar{\mu}_{B^\star}$ completes the proof. $\qquad\square$

