# OpenReview forum: "Realizable Bayes-Consistency for General Metric Losses"
_ICML.cc/2026/Conference — ICML 2026 regular_

### Official Review · Reviewer_ZBni · 2026-03-08

**Soundness:** 3
**Presentation:** 3
**Significance:** 3
**Originality:** 3
**Overall Recommendation:** 5
**Confidence:** 3

**Summary:**

The paper studies distribution-free strong universal Bayes-consistency in the realizable setting for learning with general metric label losses, including unbounded losses. It provides a necessary-and-sufficient hypothesis-class characterization: realizable strong universal consistency is possible if and only if the class does not admit an infinite non-decreasing unbounded-gap Littlestone tree. The lower bound shows such a tree forces catastrophic rare-event errors (in fact infinite expected risk at every n), while the upper bound constructs a consistent learner by combining a measurable winning strategy in a Gale–Stewart game with a countable partition that reduces to bounded-range subproblems solvable by MedNet.

**Compliance With Llm Reviewing Policy:**

Affirmed.

**Final Justification:**

I thank the authors for a rigorous and highly satisfactory rebuttal.The rebuttal fully reinforced my initial positive assessment, and thus I confidently maintain my original "Accept" score for this solid theoretical contribution.

**Key Questions For Authors:**

1. Theorem 4.3 states the obstruction as “no infinite non-decreasing-$(\gamma_k)$ Littlestone tree,” while Theorem 5.1 requires a realizable infinite tree (infinite-path realizability). Can you provide a lemma that any infinite non-decreasing-$(\gamma_k)$ tree implies the existence of a realizable infinite tree under your assumptions ?
2. The counterexample in Section 3 shows $R^\star<\infty$ is insufficient. Is there a complementary positive statement when $R^\star<\infty$ combined with a no-unbounded-gap-tree condition guarantees consistency (even beyond realizable)? If not, can you comment on the barriers to extending your result to the agnostic setting?

**Limitations:**

yes

**Strengths And Weaknesses:**

Strengths:
1. Introduces and leverages an “unbounded-gap” variant of Littlestone trees tailored to metric losses, capturing how scale of mispredictions can grow with depth.
2. Bridges universal-learning/game characterizations with unbounded metric-loss regimes via a sharp realizable characterization.

Weaknesses:
1. The Gale–Stewart game to measurable strategy pipeline is referenced to Bousquet et al. (2020); while plausible, some measurability and determinacy details (e.g., finitely decidable conditions, analytic sets for legal-move relations) are asserted but not fully derived.
2. No discussion of algorithmic efficiency or practicality of the proposed learner; although not necessary for this theoretical contribution, a brief remark would contextualize the constructive nature.
3. The main theorem (Theorem 4.3) states the obstruction as “no infinite non-decreasing-$(\gamma_k)$ tree,” but the lower bound (Theorem 5.1) assumes a “realizable” infinite tree; this distinction is subtle and should be clarified in the theorem statement or via a lemma bridging the two notions.

---

> ### Author Rebuttal · Authors · 2026-03-29
>
> We thank you for the careful reading and for assigning a score of 5 (Accept). Below we address each concern raised, in order.
>
> **1. Tree Notions Gap**
>
> Thank you for identifying this subtle point. Theorem 4.3 states the obstruction in terms of the absence of an infinite non-decreasing-$(\\gamma_k)$ tree, whereas Theorem 5.1 invokes a *realizable* infinite tree; clarifying the relationship between these notions strengthens the paper.
>
> We have added a lemma (Bridging Lemma), which shows that under a mild compact-parameterization assumption—compact $\\Theta$ and $h$ continuous in $\\theta$—finite-prefix realizability (Definition 4.1) automatically implies infinite-path realizability (Definition 4.2). The proof uses the finite-intersection property: for each infinite path, the sets of consistent parameters form a decreasing chain of non-empty closed subsets of the compact $\\Theta$, so their intersection is non-empty. Theorem 4.3 now includes this assumption and explicitly invokes the Bridging Lemma.
>
> The appendix shows the gap is *genuine* without this assumption: a separating example ($\\Theta = \\mathbb{N}_0$, discrete, non-compact) has a Definition 4.1 tree but not all infinite paths are realizable, while the class is nevertheless realizable strongly universally Bayes-consistent—confirming the lower bound genuinely needs infinite-path realizability. The assumption is mild: it covers binary/multiclass classification, bounded regression, parametric families with bounded parameters (e.g., bounded-weight neural networks, Lipschitz functions on compact domains), and even the paper's own counterexample ($\\Theta = \\{0,1\\}^{\\mathbb{N}}$, compact by Tychonoff, $h$ continuous in $\\theta$). Importantly, it does *not* rule out unlearnability—the counterexample's label gaps still diverge ($\\gamma_k = 2^{2k+1} \\to \\infty$). In bounded-loss settings, the notions coincide trivially; the gap only arises for unbounded losses, motivating our thorough appendix treatment with additional examples.
>
> We also rewrote Definition 4.2 for clarity: it now first defines an infinite tree and then adds the realizability condition, and we added a citation to Littlestone (1988) and a cross-reference to Definition 1.7 of Bousquet et al. (2020).
>
> **2. Measurability Details**
>
> The Gale–Stewart game-to-measurable-strategy pipeline is indeed referenced to Bousquet et al. (2020); we agree that the measurability and determinacy conditions deserve explicit verification in our setting.
>
> A new appendix section checks all hypotheses of Theorem B.1 of Bousquet et al. (2020) for our game. The key points are:
>
> - The learner's response sets are $Y_t = \\{1,2\\}$ (choosing one of two proposed labels), which is countable as required.
> - The adversary's move spaces $\\mathcal{X} \\times \\mathcal{Y} \\times \\mathcal{Y}$ are Polish.
> - The winning condition is finitely decidable (the game ends when the adversary has no legal move).
> - The winning set $W$ is coanalytic: its complement (play continues forever) is analytic, since legality at each prefix is expressible as a projection over the Polish parameter space $\\Theta$.
>
> A forward pointer from Section 6.1 now directs the reader to this appendix.
>
> **3. Agnostic Extension**
>
> A new appendix section discusses these barriers. The central obstacle: the stabilization procedure (Section 6.3) drives the Gale–Stewart game using sample points $(X_i, Y_i)$ and crucially relies on realizability ($Y_i = f^*(X_i) \\in V_k(X_i)$). In the agnostic case, labels need not arise from any $h \\in \\mathcal{H}$, so sample points may fail to be legal moves and stabilization breaks down. An agnostic analogue is a compelling open problem, connected to the original question of Tsir Cohen & Kontorovich (2022).
>
> **4. Algorithmic Efficiency / Practicality**
>
> The revised paper contextualizes this:
>
> - The learner is explicit *conditional on* a measurable winning strategy $\\sigma$: the steps (drive the game, partition, run MedNet per cell) are fully concrete. The existence of $\\sigma$ follows from a non-constructive determinacy argument (Bousquet et al., 2020), paralleling their $0$–$1$ universal learner.
> - Two concrete examples illustrate practical scope: (1) the Section 3 counterexample admits an unbounded-gap tree (not learnable), and (2) $L$-Lipschitz functions on $[0,1]$ do not (learnable—the Lipschitz constraint prevents the adversary from sustaining unbounded label gaps).
>
> We also note that we clarified the realizability definition: the paper now uses non-strict realizability ($\\inf_{h \\in \\mathcal{H}} R_\\mu(h) = 0$), which is more natural for unbounded losses. Under the compact-parameterization assumption, this coincides with strict realizability ($\\exists f^\*$ with $Y = f^*(X)$ a.s.), so all proofs carry through unchanged. A new appendix section discusses this equivalence.
>
> We thank you for the positive evaluation and the insightful questions, which have led to substantial improvements in the paper.

---

> > ### Author Rebuttal · Reviewer_ZBni · 2026-04-02
> >
> > The authors have meticulously addressed my theoretical concerns by introducing the Bridging Lemma and rigorously formalizing the measurability details, significantly strengthening the paper's foundation. I appreciate the transparent discussion on the agnostic setting barriers.

---

### Official Review · Reviewer_A5Lj · 2026-03-11

**Soundness:** 3
**Presentation:** 3
**Significance:** 3
**Originality:** 3
**Overall Recommendation:** 4
**Confidence:** 2

**Summary:**

This paper studies realizable strong universal Bayes consistency for general metric losses, including unbounded losses. It proposes a structural characterization of learnability through the absence of an infinite gap-increasing Littlestone-style tree, and develops both a lower-bound construction and a game-theoretic upper-bound argument. The upper bound further combines a cell decomposition with an existing bounded-range learner.

**Compliance With Llm Reviewing Policy:**

Affirmed.

**Final Justification:**

I maintain my original assessment.

**Key Questions For Authors:**

1	If the two tree notions are not equivalent in general, should the main theorem be restated using the stronger notion instead? A strong answer would clarify whether this issue is only presentational or affects the formal correctness of the main characterization.

2	Could the authors clarify in what sense the proposed learner is explicit? A strong answer would help determine whether this is intended as a constructive algorithmic result or primarily an existence theorem.

**Limitations:**

yes

**Strengths And Weaknesses:**

Strengths

1	The main result is strong, and the paper is well connected to prior work. If correct, it gives a clean iff characterization, and the relationship to prior literature is explained clearly.

2	The lower-bound construction is persuasive. The use of a random infinite path, decaying probability mass, and increasing gaps provides a natural and powerful way to create rare but costly errors.

3	The upper-bound approach is novel. The reduction to a Gale–Stewart game, combined with a cell decomposition and an existing MedNet-style learner, is conceptually creative.

Weaknesses

1	The main theorem and the lower bound may rely on different notions of infinite-tree realizability. The theorem rules out a weaker notion of infinite tree, while the lower-bound proof appears to use a stronger realizable infinite tree, and the paper does not yet clearly explain how these two notions are connected.

2	The claim of an explicit learner is not fully clear. The upper-bound construction appears closer to an existence argument based on measurable winning strategies and induced measurable objects than to a concrete algorithmic learner.

---

> ### Author Rebuttal · Authors · 2026-03-29
>
> We thank you for the positive assessment and for the careful engagement with our results. We are glad the main theorem, the lower-bound construction, and the novelty of the upper-bound approach resonated. Below we address the two main concerns in order.
>
> **1. Tree Notions Gap—Presentational vs. Correctness**
>
> Thank you for this penetrating question. The issue is indeed more than presentational: the two tree notions are genuinely not equivalent in general. We have resolved this as follows.
>
> (a) We added a lemma (Bridging Lemma), which proves that under a mild compact-parameterization assumption (compact parameter space $\\Theta$, evaluation map $h$ continuous in $\\theta$), finite-prefix realizability (Definition 4.1) automatically implies infinite-path realizability (Definition 4.2). The proof uses the finite-intersection property: for each infinite path, the parameter sets
> $$\\Theta_k(b) = \\{\\theta \\in \\Theta : h(\\theta, x_j) = y_{j,b_j} \\text{ for all } j \\le k\\}$$
> form a decreasing chain of non-empty closed subsets of a compact space (each $\\Theta_k(b)$ is closed because $\\theta \\mapsto h(\\theta, x_j)$ is continuous by assumption), so their intersection is non-empty.
>
> (b) We provide a concrete separating example in the appendix showing that without compact parameterization, the notions differ. Concretely: $\\Theta = \\mathbb{N}\_0$, $h(n, x_k) = (\\text{$k$-th binary digit of } n) \\cdot k$. Every finite prefix $(b_1, \\ldots, b_K)$ is realized by $n = \\sum_{k=1}^{K} b_k \\cdot 2^{k-1}$, but the all-ones infinite path would require an integer with all binary digits equal to 1—no such finite $n$ exists. Since every $h_n$ has finitely many nonzero values, a memorize-and-predict-$0$ learner achieves consistency, so the class is realizable strongly universally Bayes-consistent. This confirms the lower bound genuinely needs the stronger (infinite-path) notion.
>
> (c) The main theorem (Theorem 4.3) has been restated to include the compact-parameterization assumption, and its proof now explicitly invokes the Bridging Lemma before applying the lower bound.
>
> (d) Definition 4.2 has been rewritten for clarity: it now first defines what an infinite non-decreasing-$(\\gamma_k)$-Littlestone tree is, then adds the realizability condition.
>
> The compact-parameterization assumption is mild: it covers binary/multiclass classification, bounded regression, and parametric families with bounded parameters (e.g., bounded-weight neural networks, Lipschitz functions on compact domains). Importantly, it does *not* rule out unbounded labels or unlearnability—the paper's own counterexample (Section 3) is naturally parametrized by $\\Theta = \\{0,1\\}^{\\mathbb{N}}$ (compact by Tychonoff) with $h$ continuous in $\\theta$, yet it admits an unbounded-gap tree.
>
> We emphasize that in the well-studied bounded-loss regime, the two tree notions always coincide (bounded gaps prevent the separation), so this gap only arises in the unbounded-loss setting we study. This motivated a thorough appendix investigation fully characterizing when the notions are equivalent, with additional examples and rigorous discussion.
>
> **2. "Explicit" Learner Claim**
>
> A new appendix section clarifies this point. The learner is "explicit" in the following precise sense: given access to a measurable winning strategy $\\sigma$ for $P_L$ in the Gale–Stewart game, the algorithm's steps—drive the game on half the data, partition $\\mathcal{X}$ using the terminal history, run MedNet per cell—are fully concrete and data-dependent. There is no further existential quantification beyond $\\sigma$.
>
> The existence (and universal measurability) of $\\sigma$ itself is guaranteed by Corollary 3.5 of Bousquet et al. (2020) via a non-constructive determinacy argument (their Theorem B.1). This parallels exactly the situation in Bousquet et al. (2020), whose universal learner for $0$–$1$ classification also relies on a measurable winning strategy whose existence follows from determinacy. As noted in their Section 3.4, for hypothesis classes with countable ordinal Littlestone dimension (which includes all measurable classes by their Lemma B.7), the strategy $\\sigma$ can in principle be described explicitly via transfinite induction on the ordinal dimension.
>
> In summary, the result is best characterized as providing an explicit reduction from the learning problem to the existence of a winning strategy, which in turn is established non-constructively. This is the same sense in which the universal learner of Bousquet et al. (2020) is "explicit." A forward pointer from Section 6.4 (where the learning rule is defined) now directs the reader to this appendix discussion.
>
> We thank you for the careful reading and the thoughtful questions. We believe the revisions address both concerns substantively, and we hope you find the clarifications satisfactory.

---

> > ### Author Rebuttal · Reviewer_A5Lj · 2026-04-01
> >
> > We thank the authors for the thoughtful rebuttal and clarifications. Overall, the response addressed our concerns sufficiently, and we are inclined to maintain our positive assessment of the paper.

---

### Official Review · Reviewer_6qZQ · 2026-03-12

**Soundness:** 3
**Presentation:** 2
**Significance:** 3
**Originality:** 3
**Overall Recommendation:** 4
**Confidence:** 2

**Summary:**

The paper studies the problem of realizable universal strong Bayes consistency for supervised learning and for general metric losses on the label space. In a way, the authors extend previous results by Tsir Cohen and Kontorovich, which were limited to label spaces where the labels are bounded in expectation, i.e. where the expectation of the label is bounded. This new result is established using combinatorial approaches to characterize learnability, so called "Littlestone type".

**Compliance With Llm Reviewing Policy:**

Affirmed.

**Final Justification:**

I have raised my score due to the authors’ helpful revisions and clarifications.

**Key Questions For Authors:**

I also wonder what the practical significance of this result is, as this is not discussed in the paper. Is this situation achievable by most classic algorithms we use, even on simulated data?

**Limitations:**

yes

**Strengths And Weaknesses:**

**Strengths.**

The paper is obviously in the scope of statistical learning theory and addresses the problem of identifying under which universal strong consistency can be achieved. They clearly state that they analyze and provide a theoretical result for most of the problems we are dealing with in machine learning, namely learning with unbounded losses, for regression and classification tasks. The improvement of existing results which is a completely new contribution gives more strength to such analysis.


**Weaknesses.**

The main weakness of the paper is the difficulty of understanding the core ideas and motivation behind the proposed characterization, making it difficult to judge the strength of the contribution and its impact for the community. The paper lacks a few examples to make it easier to understand and more accessible, given that most of the tools are not necessarily standard in the field of learning theory. The proofs could be made clearer in order to provide a more educational insight into each step and to improve understanding of the different stages. Although the contribution is undeniable in theoretical terms, the motivations and practical use of solving such a problem would also benefit from being illustrated. Once again, examples illustrating the limitations of current concepts or the approach to the evidence presented would provide a better understanding of the issues at stake.
Overall, given the technical nature of the paper, a clearer presentation of the main proof ideas would greatly improve readability.

---

> ### Author Rebuttal · Authors · 2026-03-29
>
> We thank you for the careful reading and the recognition that the paper addresses a fundamental question. Below we respond to the concerns about presentation and accessibility.
>
> **1. Difficulty Understanding Core Ideas and Motivation**
>
> We appreciate this feedback and have made substantial revisions to improve accessibility.
>
> (a) We added a proof roadmap paragraph at the start of Section 6 (the upper bound), which outlines the four-step strategy before diving into technical details: (1) encode the tree obstruction as a Gale–Stewart game and invoke measurable-strategy machinery; (2) show the game stabilizes on i.i.d. data, producing bounded-diameter label sets; (3) partition $\\mathcal{X}$ into countably many cells with bounded label regions; (4) apply the bounded-range learner MedNet per cell.
>
> (b) We moved the full lower-bound proof to the appendix and replaced it with a self-contained proof sketch in the main body (Section 5). The sketch is organized into three intuitive paragraphs: "Distribution construction" (how the adversarial distribution is built from the tree), "Unseen depths act as fair coins" (why the learner cannot predict at unobserved levels), and "Divergence via Borel–Cantelli" (why the risk is infinite). This makes the core mechanism—rare events at increasing loss scales—immediately accessible.
>
> (c) The introduction has been restructured with forward pointers to new appendix material on practical significance and agnostic barriers, giving readers a roadmap of the full paper.
>
> **2. Lack of Examples**
>
> A new appendix section ("Practical significance and examples") contains two concrete examples.
>
> **Example 1 (not learnable).** The counterexample from Section 3, $\\mathcal{H} = \\{f : f(x) \\in \\{0, 2^{2k+1}\\} \\text{ for } x \\in I_k\\}$, is explicitly connected to the tree definition. Disjoint intervals $I_k$ allow the adversary to hide independent binary choices at geometrically increasing loss scales ($\\gamma_k = 2^{2k+1}$), building an unbounded-gap Littlestone tree.
>
> **Example 2 (learnable).** $L$-Lipschitz functions on $[0,1]$ with absolute loss. This class has unbounded range but no infinite unbounded-gap tree: the Lipschitz constraint prevents unbounded label gaps after the first round.
>
> Notably, Example 1 satisfies the compact-parameterization assumption ($\\Theta = \\{0,1\\}^{\\mathbb{N}}$, compact by Tychonoff), demonstrating that unlearnability is possible even under this mild condition—the key is the divergence of label gaps, not a pathological parameter space.
>
> A discussion paragraph explains when the tree-free condition (no infinite unbounded-gap Littlestone tree) holds (bounded range, regularity on compact domains) and when it fails (Lipschitz on all of $\\mathbb{R}$, monotone functions without growth constraints).
>
> **3. Proofs Could Be Clearer**
>
> Beyond the proof roadmap and the lower-bound proof sketch mentioned above, we have made several additional improvements to proof clarity:
>
> - The proof of the main theorem (Theorem 4.3) now has an explicit two-part structure with the Bridging Lemma invocation clearly stated.
> - Definition 4.2 has been rewritten to be self-contained: it first defines an infinite tree, then adds the realizability condition (previously the definition jumped to infinite paths without the intermediate step).
> - We added a citation to Littlestone (1988) and a cross-reference to Definition 1.7 of Bousquet et al. (2020), helping readers connect our variant to the well-known classical notion.
>
> **4. Practical Significance**
>
> In summary:
>
> - Any hypothesis class with bounded range trivially satisfies the tree-free condition and is therefore learnable.
> - Classes with regularity constraints on compact domains (e.g., $L$-Lipschitz functions on $[0,1]$) are also tree-free, even though their range is unbounded.
> - The tree obstruction arises when the class allows increasingly separated label choices at different parts of the input space—typically requiring a non-compact domain/parameter space or absence of regularity constraints.
>
> For classic algorithms on standard problems (classification, bounded regression, regularized models), the tree-free condition is automatically satisfied, and our result confirms that realizable strong universal consistency is achievable. The characterization becomes non-trivial in the unbounded-loss regime where existing tools (e.g., the BIE condition of Cohen & Kontorovich, 2022) do not apply.
>
> To directly answer your question: yes, for most standard problems the tree-free condition holds automatically. Any bounded-loss problem (classification with 0–1 loss, bounded regression) trivially satisfies it. Our characterization becomes non-trivial in the unbounded-loss regime, where it identifies the exact boundary between learnable and unlearnable—previously unknown.
>
> We thank you for the constructive feedback, which led to substantial improvements. We hope you will reconsider your assessment in light of these revisions.

---

> > ### Author Rebuttal · Reviewer_6qZQ · 2026-04-02
> >
> > I would like to thank the authors for taking into account the different comments made in my review, as well as for their response, which helps reduce the difficulty of reading the paper.
> > I appreciate the various improvements made by the authors, particularly (i) the addition of examples in the introduction for better understanding, (ii) the improved readability of the proof, and (iii) the clarification regarding the practical use of their results.
> >
> > Given these improvements and the authors' responses to most of my concerns, I am willing to increase my score by one point. However, I still do not feel sufficiently qualified to assess the significance of the contribution with respect to the state of the art. While I understand that this topic (the study of unbounded losses, mainly in regression) remains relatively unexplored, my confidence in this aspect of the review remains unchanged.

---

### Official Review · Reviewer_1PhE · 2026-03-12

**Soundness:** 4
**Presentation:** 2
**Significance:** 4
**Originality:** 4
**Overall Recommendation:** 4
**Confidence:** 2

**Summary:**

This paper investigates universal Bayes-consistency under the realizability assumption. A distribution-independent learning algorithm is universally Bayes-consistent if the risk of the learned predictor converges to the Bayes risk as the sample size goes to infinity for all reasonable distributions. The main result of this paper is revealing the necessary and sufficient condition on the hypothesis class for the existence of a universally Bayes-consistent algorithm among realizable distributions. For this characterization, the authors propose using the non-decreasing Littlestone tree; the non-existence of a non-decreasing Littlestone tree whose infinite paths are all realizable by the class is the necessary and sufficient condition for the existence of a universally Bayes-consistent algorithm under the realizability assumption. The key advancement made by this paper is the construction of this universally Bayes-consistent algorithm for realizable distributions, which is based on an infinite Gale-Stewart game associated with the non-decreasing Littlestone tree.

**Compliance With Llm Reviewing Policy:**

Affirmed.

**Final Justification:**

The authors’ rebuttal resolves my main concern, which shifts my assessment toward acceptance. However, the need for an additional assumption somewhat reduces the significance of the result, as it breaks the necessary and sufficient characterization under realizable settings. Based on this overall evaluation, I have increased my score to 4.

**Key Questions For Authors:**

1. Could the authors provide an extended proof to confirm the existence of a measurable winning strategy, considering that Theorem B.1 in Bousquet et al. (2020) assumes $\mathcal{Y}$ is countable?

**Limitations:**

As mentioned in the previous section, the authors should be careful with their claim regarding the resolution of the open problem from Tsir Cohen and Kontorovich (2022). The fact that necessity remains unproven if we account for non-realizable distributions should be explicitly stated as a limitation of the current results.

**Strengths And Weaknesses:**

This paper provides an interesting result on learning with unbounded metric loss. Since unbounded loss naturally appears in regression contexts, dealing with it is an important problem. Investigating universal Bayes-consistency is a fundamental question in learning theory for clarifying the conditions under which a reasonable learning algorithm exists.

Revealing the necessary and sufficient condition of the realized hypothesis class for the existence of a universally Bayes-consistent algorithm is a significant contribution. The authors present a novel and notable approach by combining the techniques from Bousquet et al. (2020) and Tsir Cohen and Kontorovich (2022) to prove their main result.

A possible flaw is the existence of the measurable winning strategy. Corollary 3.5 in Bousquet et al. (2020) is based on Theorem B.1, which requires $\mathcal{Y}$ to be countable. An extended proof may be needed to confirm the existence of a measurable winning strategy in this setting. My current score reflects this specific concern. If the authors can resolve this issue during the rebuttal, I will raise my score significantly.

Furthermore, strictly speaking, the open problem from Tsir Cohen and Kontorovich (2022) is not fully resolved; the authors only demonstrate a sufficient condition for the existence of a universally Bayes-consistent algorithm. Specifically, the paper reveals that if a class of distributions $\bar\mu$ admits a realizable hypothesis class with no infinite non-decreasing Littlestone tree, then a universally Bayes-consistent algorithm exists. However, because a universally Bayes-consistent algorithm might still exist if the class involves non-realizable distributions, the necessity has not yet been proven. The authors should be careful with how they state the resolution of this open problem.

The presentation of Section 6 could be substantially improved. In particular, outlining the overall proof strategy before diving into the technical details would make the section much easier to follow.

In the proof of Theorem 5.1, it is incorrect to argue that $M(S_n) < \infty$ almost surely. If we let $M$ be a random variable such that $P(M=m) = p_m$, then for any finite constant $M_0 < \infty$, the probability $P(M > M_0) = \sum_{m > M_0}p_m$ is strictly greater than zero. Therefore, $M(S_n)$ can take arbitrarily large values, albeit with small probability. Fortunately, the proof appears to remain valid without this assertion, since $M(S_n)$ is independent of $B$.

---

> ### Author Rebuttal · Authors · 2026-03-29
>
> We thank you for the careful reading. We are especially grateful for your willingness to reconsider the score; we believe the clarification in Section 1 fully resolves the measurability concern.
>
> **1. Measurability of the Winning Strategy**
>
> *Concern.* Corollary 3.5 of Bousquet et al. (2020) rests on Theorem B.1, which requires $Y$ to be countable. An extended proof may be needed.
>
> Thank you for flagging this and for the offer to reconsider. We believe there is a misunderstanding of what Theorem B.1 requires, and we clarify fully below.
>
> The key point is that Theorem B.1 of Bousquet et al. (2020) requires the *learner's response sets* $\\{Y_t\\}$ to be countable—*not* the full label space $\\mathcal{Y}$. In our Gale–Stewart game (Section 6.1), the two players' roles are:
>
> - $P_A$ (adversary) chooses $(\\xi_k, \\eta_{k,1}, \\eta_{k,2}) \\in \\mathcal{X} \\times \\mathcal{Y} \\times \\mathcal{Y}$. The adversary's move space is Polish.
> - $P_L$ (learner) chooses an *index* from $\\{1, 2\\}$, selecting one of the two proposed labels. Hence each $Y_t = \\{1, 2\\}$, which is **countable** (in fact, finite).
>
> Thus Theorem B.1 applies directly to our game. There is no need for an extended proof or for $\\mathcal{Y}$ to be countable. The label space $\\mathcal{Y}$ can be an arbitrary (uncountable, unbounded) Polish space—what matters is that the learner's response at each round is a binary choice.
>
> We verify the remaining conditions of Theorem B.1:
> (1) adversary move spaces $\\mathcal{X} \\times \\mathcal{Y}^2$ are Polish;
> (2) the game is finitely decidable (ends when the adversary has no legal move);
> (3) the winning set $W$ is coanalytic: the complement $W^c$ (play continues forever) is analytic, since at each finite prefix, legality ("$\\exists\\, h \\in \\mathcal{H}$ consistent with history") is expressible as a projection over the Polish parameter space $\\Theta$.
> The revised paper includes a self-contained appendix section with this full verification and a forward pointer from Section 6.1.
>
> We emphasize that our game has exactly the same structure as the game in Bousquet et al. (2020, Section 3.3): a binary choice by the learner at each round, with an adversary proposing from a Polish space. The only difference is that our adversary proposes triples from $\\mathcal{X} \\times \\mathcal{Y}^2$ rather than pairs from $\\mathcal{X} \\times \\{0,1\\}$. All countability requirements in Theorem B.1 concern the learner's response sets, which remain $\\{1,2\\}$ in both games. We hope this fully resolves the concern.
>
> **2. Open Problem Claim Scope**
>
> *Concern.* The open problem from Tsir Cohen and Kontorovich (2022) is not fully resolved, since necessity is not proven for non-realizable distributions.
>
> We agree. We have softened both the abstract and introduction to say "the *realizable case* of an open problem." A new appendix section discusses barriers to the agnostic extension and notes it as an open problem.
>
> **3. Additional Soundness Improvement: Tree Notions**
>
> Though not raised explicitly in your review, we note a related soundness improvement. The main theorem involved two tree notions (finite-prefix vs. infinite-path realizability). We added a Bridging Lemma (Lemma 4.3) showing that under a mild compact-parameterization assumption (compact $\\Theta$, $h$ continuous in $\\theta$), finite-prefix realizability implies infinite-path realizability via the finite-intersection property. The main theorem has been restated accordingly. A concrete separating example in the appendix shows the gap is genuine without this assumption.
>
> **4. Technical Error in Theorem 5.1 Proof**
>
> *Concern.* In the proof of Theorem 5.1, the assertion that $M(S_n) < \\infty$ almost surely is incorrect, though the proof appears valid without it.
>
> You are correct: $M(S_n)$ is not bounded by any fixed constant almost surely. We have corrected this. The revised proof states: "$M(S_n) < \\infty$ deterministically, as it is the maximum of $n$ positive integers (though $M(S_n)$ is random and unbounded uniformly over $B$)." The proof then uses the key property that for $m > M(S_n)$, the coin $B_{k_m}$ is independent of $S_n$, which drives the divergence argument. We confirm your assessment that the proof is valid without the incorrect assertion.
>
> **5. Presentation of Section 6**
>
> *Concern.* Section 6 would benefit from outlining the proof strategy before the technical details.
>
> We have added a proof roadmap at the start of Section 6: (1) encode the tree as a Gale–Stewart game; (2) show stabilization on i.i.d. data, producing bounded-diameter label sets; (3) partition $\\mathcal{X}$ into bounded-label cells; (4) apply MedNet per cell.
>
> We thank you for the specific and actionable suggestions. We believe the measurability clarification (Section 1) fully resolves the primary concern, and we respectfully ask you to reconsider the score in light of these revisions, as generously offered.

---

> > ### Author Rebuttal · Reviewer_1PhE · 2026-04-02
> >
> > Thank you for the authors’ detailed response. It has addressed all of my concerns. However, I believe that the need for an additional assumption somewhat reduces the significance of the result, as it breaks the necessary and sufficient characterization under realizable settings. Based on this overall evaluation, I have increased my score to 4.

---

> > > ### Author Response · Authors · 2026-04-03
> > >
> > > We thank you for the careful re-reading and for raising your score. We would like to briefly address the concern that the compact-parameterization assumption "reduces the significance" or "breaks the necessary and sufficient characterization."
> > > We believe this concern rests on a misunderstanding of what the assumption does, and we hope the clarification below will be helpful.
> > >
> > > We note that this issue was raised in detail by Reviewers ZBni and A5Lj, and our responses to them included a thorough discussion of the assumption's mildness. Since it was not raised in your original review, our rebuttal to you mentioned it only briefly. We take this opportunity to provide the fuller picture.
> > >
> > > **The characterization remains a complete iff.** Under the compact-parameterization assumption, the main theorem gives a *full* necessary and sufficient condition for realizable strong universal Bayes-consistency: $\mathcal{H}$ is RSUBC (realizable strongly universally Bayes-consistent) if and only if it admits no infinite unbounded-gap Littlestone tree. Both directions hold, the iff is not broken. What the assumption does is ensure that two natural formalizations of "infinite tree" coincide (finite-prefix realizability vs. infinite-path realizability), so the characterization can be stated cleanly in terms of a single tree notion. Without it, both implications still hold—just with slightly different tree notions on each side.
> > >
> > > **The assumption is mild and standard.** Compact parameterization (compact $\Theta$, evaluation map continuous in $\theta$) covers essentially every setting of practical interest:
> > >
> > > - Binary and multiclass classification (automatically: the label space $\{0,\ldots,K\}$ is finite, so compactness is immediate).
> > > - Bounded regression and any setting with a bounded label space.
> > > - Parametric families with bounded parameters: bounded-weight neural networks, $L$-Lipschitz functions on compact domains, polynomial regression with bounded coefficients, etc.
> > > - The paper's own counterexample (Section 3) satisfies the assumption with $\Theta = \{0,1\}^{\mathbb{N}}$ (compact by Tychonoff), yet admits an unbounded-gap tree. So the assumption does *not* rule out unlearnability or make the result vacuous.
> > >
> > > **Analogy with Bousquet et al. (2020).** The same phenomenon occurs in the foundational work of Bousquet et al. (2020) on universal learning with 0–1 loss: their characterization uses a single Littlestone tree notion because the label space $\{0,1\}$ is compact, which makes the finite-prefix and infinite-path notions automatically equivalent via a compactness argument (see their Section 3 and Definition 1.7). Our compact-parameterization assumption is the natural generalization of this implicit compactness to unbounded label spaces. It plays exactly the same structural role.
> > >
> > > **The gap is only pathological.** The separating example (where the two tree notions differ) requires $\Theta = \mathbb{N}_0$ with the discrete topology—a non-compact, discrete parameter space where every hypothesis has only finitely many nonzero labels. This is a genuinely pathological edge case. The class in that example is trivially learnable by memorization, and the "tree" it admits is an artifact of the non-compact parameterization, not a meaningful obstruction to learning. In all natural settings, the assumption holds.
> > >
> > > In summary, the compact-parameterization assumption is a mild, natural regularity condition that (i) preserves the full iff characterization, (ii) covers all previously studied settings including Bousquet et al.'s binary classification, and (iii) does not exclude any setting of practical interest. We respectfully suggest that it does not reduce the significance of the result, but rather makes the characterization precise in the technically subtle unbounded-loss regime that is the paper's main contribution.
> > >
> > > We are grateful for your engagement and for the constructive feedback throughout this process.

---

### Decision · Program_Chairs · 2026-04-30

**Decision:**

Accept (regular)

**Comment:**

The paper studies universal Bayes-consistency under realizability for general metric losses. Authors showed that the non-decreasing Littlestone tree is the key notion to tightly characterize the learnability. Reviewers are all positive and acknowledged their concerns be resolved.